# Emotion-LLaMA: Multimodal Emotion Recognition and Reasoning with Instruction Tuning

**Zebang Cheng**[12*]    **Zhi-Qi Cheng**[3*†]  **Jun-Yan He**[4]  **Jingdong Sun**[3]

**Kai Wang**[5]  **Yuxiang Lin**[1]  **Zheng Lian**[6]  **Xiaojiang Peng**[12†] **Alexander G. Hauptmann**[3]

[1]Shenzhen Technology University [2]Shenzhen University [3]Carnegie Mellon University [4]Alibaba Group
[5]National University of Singapore    [6]Institute of Automation, Chinese Academy of Sciences

**Project:** https://zebangcheng.github.io/Emotion-LLaMA
**Demo:** https://huggingface.co/spaces/ZebangCheng/Emotion-LLaMA

## Abstract

Accurate emotion perception is crucial for various applications, including human-computer interaction, education, and counseling. However, traditional single-modality approaches often fail to capture the complexity of real-world emotional expressions, which are inherently multimodal. Moreover, existing Multimodal Large Language Models (MLLMs) face challenges in integrating audio and recognizing subtle facial micro-expressions. To address this, we introduce the *MERR dataset*, containing 28,618 coarse-grained and 4,487 fine-grained annotated samples across diverse emotional categories. This dataset enables models to learn from varied scenarios and generalize to real-world applications. Furthermore, we propose *Emotion-LLaMA*, a model that seamlessly integrates audio, visual, and textual inputs through emotion-specific encoders. By aligning features into a shared space and employing a modified LLaMA model with *instruction tuning*, Emotion-LLaMA significantly enhances both emotional recognition and reasoning capabilities. Extensive evaluations show Emotion-LLaMA outperforms other MLLMs, achieving top scores in Clue Overlap (7.83) and Label Overlap (6.25) on EMER, an F1 score of 0.9036 on MER2023-SEMI challenge, and the highest UAR (45.59) and WAR (59.37) in zero-shot evaluations on DFEW dataset.

## 1  Introduction

Emotion perception plays a vital role in applications such as human-computer interaction [17–19, 80], educational assistance [43], and psychological counseling [7, 42]. While single-modality approaches, including *facial expression recognition* [45, 57, 70, 87], *text emotion analysis* [25, 49, 41], and *audio emotion recognition* [31, 39, 47], have shown effectiveness, real-world emotional data is often multimodal, integrating text, audio, and images.

Despite extensive multimodal fusion methods having achieved promising improvements [13, 14, 16, 56, 58, 59, 77, 91, 96, 100, 103], they mainly focus on feature interaction and modality completion, remaining under-explored for knowledge-level interaction which is essential for emotional reasoning of humans. Recently, *Multimodal Large Language Models (MLLMs)* have excelled in tasks such as visual-language understanding [50, 65], visual question answering [104], and video understanding [35, 63, 86]. However, for emotion recognition [45], models like GPT-4 with Vision (GPT-4V) still face two main challenges: the inability to process audio and the failure to recognize micro-expressions.

---

*Equal contribution, listed in random order, completed during Zebang Cheng's internship at CMU.
†Corresponding author (zhiqic@cs.cmu.edu, pengxiaojiang@sztu.edu.cn).

38th Conference on Neural Information Processing Systems (NeurIPS 2024).

We argue that the lack of specialized multimodal emotion instruction datasets is the main factor limiting MLLMs' effectiveness. These issues stem from the inability of previous methods to effectively integrate audio inputs, which are crucial for capturing vocal tones and auditory cues, and their difficulty in recognizing subtle facial micro-expressions. These limitations lead to sub-optimal performance in real-world scenarios.

To address these challenges, we introduce the *MERR dataset* (Sec. 3.1), which enables multimodal large models and supports *instruction tuning* to learn from diverse scenarios and generalize to real-world applications. We also propose the *Emotion-LLaMA model* (Sec. 3.2), which integrates audio, visual, and textual inputs through emotion-specific encoders. By employing *instruction tuning* (Sec. 3.3), *Emotion-LLaMA* significantly enhances both the accuracy of emotional recognition and the depth of emotional reasoning, setting a new benchmark for multimodal emotion analysis. Extensive experiments and evaluations (Sec. 4) demonstrate Emotion-LLaMA's superiority, achieving top scores on EMER, MER2023[1], MER2024[2], and DFEW datasets. Our main contributions are as follows:

- We constructed the *MERR dataset*, which includes 28,618 coarse-grained and 4,487 fine-grained annotated samples, covering a wide range of emotional categories such as "doub" and "contempt". Unlike previous datasets, MERR's diverse emotional contexts allow models to learn from varied scenarios and generalize to real-world applications, serving as a valuable resource for advancing large-scale multimodal emotion model training and evaluation.

- We developed the *Emotion-LLaMA model*, which incorporates HuBERT for audio processing and multiview visual encoders (MAE, VideoMAE, EVA) for capturing facial details, dynamics, and context. By aligning these features into a modified LLaMA language model, *Emotion-LLaMA* enhances emotional recognition and reasoning capabilities.

- Extensive experiments demonstrate that *Emotion-LLaMA* significantly outperforms other MLLMs across multiple datasets, establishing it as the current state-of-the-art model in public competitions. It achieved top scores on the EMER dataset (Clue Overlap: 7.83, Label Overlap: 6.25) and attained F1 scores of 0.9036 on MER2023-SEMI[1] and 0.8452 on MER2024-NOISE[2]. *Emotion-LLaMA* also surpassed ChatGPT-4V in zero-shot evaluations, including DFEW (+4.37%) and MER2024-OV[2] (+8.52%).

## 2 Related Work

To highlight our contributions, we review existing multimodal large language models and instruction tuning methods, emphasizing their limitations in emotional understanding.

**Multimodal Large Language Models (MLLMs).** MLLMs [1, 5, 11, 20, 73, 88] have gained substantial attention due to their powerful inferential capabilities. Research primarily focuses on leveraging pretrained models like CLIP [75], Q-Former [53], and ImageBind [34] for general domain applications [10, 97, 104]. However, even advanced models like GPT-4V [61] struggle with understanding audio emotional cues and recognizing facial micro-expressions due to the lack of specialized training on multimodal emotional datasets and emotion-related knowledge. Recently, researchers have begun training MLLMs on multimodal emotional datasets to identify emotion-triggering utterances in dialogues [2, 14], though these studies often lack detailed explanations. *In contrast, our proposed Emotion-LLaMA employs emotion-specific encoders to extract multimodal features, enhancing emotional recognition and reasoning capabilities.*

**Instruction Tuning.** Language instructions have been widely used across diverse NLP tasks [6, 21, 76, 101, 95]. Studies like InstructionGPT [72], FLAN [23], and OPT-IML [44] have explored instruction-tuning methods [89, 90] that significantly enhance the zero-shot and few-shot capabilities of LLMs. The vision field has also embraced language instructions for various vision-language tasks [1, 3, 27, 40, 46, 99]. LLaVA [65] converted image-text pairs into instruction-following data using a language-only model, while EmoVIT [92] generated visual emotion instruction data using paired annotations. However, these approaches often lack audio information, which is crucial for understanding human emotions. Due to high annotation costs, AffectGPT [62] manually annotated only 100 samples with emotion clues. *To address the scarcity of emotion-related instruction-following data, our approach generates multimodal descriptions using prior knowledge.*

---

[1]http://merchallenge.cn/mer2023
[2]https://zeroqiaoba.github.io/MER2024-website

# 3 Methodology

This section presents our proposed Emotion-LLaMA model, which consists of three key components: the MERR dataset construction (Sec. 3.1), the Multimodal Emotion-LLaMA model architecture (Sec. 3.2), and the training procedures (Sec. 3.3).

## 3.1 MERR Dataset Construction

The MERR dataset is constructed through a comprehensive process of emotion annotation in video data, as outlined in Algorithm 1 and Figure 1. First, human faces are extracted from each video frame using the OpenFace toolkit, which detects and scores Action Units (AUs) [28, 69] to identify the frame with the maximum cumulative intensity:

$$\mathrm{I}_{peak} = \arg \max_k \left( \sum_i S_{au_i}^k \right), \tag{1}$$

where $S_{au_i}$ represents the intensity of each AU. These AUs are mapped to facial expression descriptions $C_{ved}$ (Tables 10 and 11) to accurately depict facial movements. Next, MiniGPT-v2[3] analyzes the peak frame to extract contextual information $C_{vod}$, such as activities and environment (Figure 1), facilitating the identification of latent emotional elements within the background context. Qwen-Audio[4] processes audio segments to extract nuances in speech and vocal tone, generating emotion-related descriptions $C_{atd}$. Visual and audio information are concatenated into a raw multimodal description, integrating sensory inputs to enhance the contextual supplementation for lexical subtitles. Lexical subtitles $C_{ls}$ are integrated into the multimodal description, providing textual context that complements the audio and visual data. LLaMA-3[5] refines these annotations by aggregating unimodal descriptions ($C_{ved}$, $C_{vod}$, $C_{atd}$, $C_{ls}$) into a detailed multimodal description $C_{md}$, following instructions and examples in Table 12. Finally, the comprehensive description $C_{md}$ is used to annotate the peak frame, ensuring the video is annotated with detailed emotional descriptors.

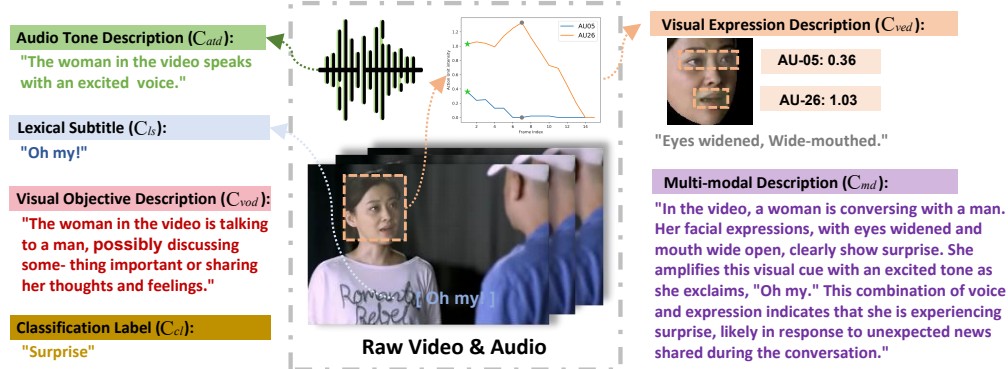

Figure 1: Example of the MERR dataset: It includes audio tone description, lexical subtitle, visual objective description, visual expression description, classification label, and multimodal description.

The MERR dataset extends the range of emotional categories and annotations beyond those found in existing datasets (Table 15). Each sample is annotated with an emotion label and described in terms of its emotional expression. The dataset was initially auto-annotated with coarse-grained labels for 28,618 samples from a large pool of unannotated data using LLaMA-3, and was later refined to include 4,487 samples with fine-grained annotations, carefully selected by experts. Figure 5 shows that, compared to other datasets, MERR encompasses a wider range of emotional categories. More details of MERR dataset construction are provided in the project homepage and the Appendix A.

## 3.2 Multimodal Emotion-LLaMA Model

The proposed Multimodal Emotion Large Language Model (Emotion-LLaMA) architecture, depicted in Figure 2, comprises an audio encoder $\mathcal{E}^{aud}$, a visual encoder $\mathcal{E}^{vis}$, and a multimodal large language

---

[3] https://github.com/Vision-CAIR/MiniGPT-4/blob/main/demo_v2.py
[4] https://www.modelscope.cn/models/qwen/QWen-Audio/summary
[5] https://huggingface.co/meta-llama/Meta-Llama-3-8B-Instruct

---
**Algorithm 1** Multimodal Emotion Annotation Procedure
---
**Require:** Video frames $V = f_1, f_2, \ldots, f_n$, Audio stream $A$
**Ensure:** Annotated video with comprehensive emotional descriptors for the peak emotional expression frame
 1: Initialize $I_{\text{peak}} \leftarrow 0$
 2: Initialize $Frame_{\text{peak}} \leftarrow \emptyset$
 3: **for** each frame $f_k$ in $V$ **do**
 4:     Detect AUs and compute $I_{\text{AU}} \leftarrow \sum_i S_{au_i}^k$
 5:     **if** $I_{\text{AU}} > I_{\text{peak}}$ **then**
 6:         $I_{\text{peak}} \leftarrow I_{\text{AU}}$
 7:         $Frame_{\text{peak}} \leftarrow f_k$
 8:     **end if**
 9: **end for**
10: Analyze $Frame_{\text{peak}}$ with OpenFace to obtain $C_{ved}$
11: Analyze $Frame_{\text{peak}}$ with MiniGPT-v2 to obtain $C_{vod}$
12: Analyze audio $A$ with Qwen-Audio to obtain $C_{atd}$
13: Integrate $C_{ls}$, $C_{ved}$, $C_{vod}$, and $C_{atd}$ to synthesize context
14: Generate comprehensive description $C_{md}$ using LLaMA-3
15: **return** $C_{md}$
---

model $\phi$. Given an input tuple $P = \langle \text{Audio}, \text{Video}, \text{Prompt} \rangle$, Emotion-LLaMA is formulated as:

$$\hat{O} = \Psi(\phi, \mathcal{E}, \Omega, P)$$
$$= \phi(\mathcal{E}^{aud}(\text{Audio}), \mathcal{E}^{vis}(\Omega(\text{Video})), \mathcal{E}^{tex}(\text{Prompt})) \quad (2)$$

where $\phi$, $\Omega$, and $\mathcal{E}$ denote the LLaMA language model [85], vision pre-processor, and multimodal encoder, respectively. $\hat{O}$ represents the formatted output text result. The multimodal encoder $\mathcal{E}$ consists of audio, vision, and text prompt encoders. Input $Video$ is pre-processed to construct the frame sequence $V$ and $Frame_{\text{peak}}$ (Sec. 3.1).

**Multimodal Prompt Template.** To address the intricate needs of emotional understanding, we craft a structured multimodal prompt template incorporating descriptive captions and emotion flags (as detailed in Table 16 and 17), directing the LLM to decipher latent correlations between emotional states and corresponding visual or auditory content. The template is denoted as:

*[INST] <VideoFeature> <AudioFeature> [Task Identifier] Prompt [/INST]*

**Multiview Multimodal Encoder.** To capture emotional cues in audio and visual modalities, we leverage the HuBERT [39] model as our audio encoder $\mathcal{E}^{aud}$ and a multiview visual encoder $\mathcal{E}^{vis}$. HuBERT extracts a comprehensive auditory representation $u^{aud}$ from the input audio signal, exhibiting remarkable performance in emotion recognition tasks.

We use a vision preprocessor to unify vision modalities, including facial sequences and peak frame extracted from the input video. Three visual encoders $\mathcal{E}^{vis} = \mathcal{E}^{vis}_{glo}, \mathcal{E}^{vis}_{loc}, \mathcal{E}^{vis}_{temp}$ are employed to comprehensively extract complementary multi-view visual emotional features:

- *Local Encoder*: A ViT-structured model pre-trained by the MAE scheme [82] extracts static facial expression features. A facial sequence is fed into the local encoder, and the output frame-wise features are fused by average pooling, producing the local visual feature $u^{vis}_{loc} = \text{AVG}(\mathcal{E}^{vis}_{loc}(V))$.

- *Temporal Encoder*: A VideoMAE [84] model, produces the temporal feature $u^{vis}_{temp} = \mathcal{E}^{vis}_{temp}(V)$ of a facial sequence, learning facial dynamics that indicate emotional states and offering a temporal dynamic view of human emotion.

- *Global Encoder*: A ViT-structured model, EVA [32], initialized with official pre-trained weights, produces the visual feature $u^{vis}_{glo} = \mathcal{E}^{vis}_{glo}(Frame_{peak})$, capturing not only facial expressions but also background context.

**Multimodal Integration and Tokenization.** We use the LLaMA tokenizer, employing a byte-pair encoding (BPE) model based on SentencePiece [48], to address open vocabulary challenges and

facilitate efficient processing of textual inputs. For multimodal emotional reasoning, a modified generate method iteratively selects the most probable tokens, producing contextually appropriate and emotionally nuanced responses.

To integrate audio and visual features with text tokens, we introduce a linear projection mechanism that transforms these features into a common dimensional space. This involves trainable linear mappings $\boldsymbol{\sigma}$, which include $\sigma^{aud}$ for the audio token, and $\sigma^{vis}_{glo}$, $\sigma^{vis}_{loc}$, and $\sigma^{vis}_{temp}$ for the visual tokens. Specifically, we apply $\boldsymbol{\sigma}$ to convert multimodal feature $u$ into language embedding tokens $\mathcal{T}$:

$$\mathcal{T} = \boldsymbol{\sigma} \cdot u, \text{ with } u = u^{aud}, u^{vis}_{glo}, u^{vis}_{loc}, u^{vis}_{temp} \tag{3}$$

The resulting multimodal tokens $\mathcal{T}$ comprise a single audio token $\langle \mathrm{T}^{aud} \rangle$, three visual tokens $\langle \mathrm{T}^{vis}_{glo} \rangle$, $\langle \mathrm{T}^{vis}_{loc} \rangle$, and $\langle \mathrm{T}^{vis}_{temp} \rangle$, and a sequence of text tokens $\langle \mathrm{T}^{tex}_0 \rangle, \dots, \langle \mathrm{T}^{tex}_N \rangle$. These tokens are fused through the inner cross-attention mechanism of Emotion-LLaMA, enabling it to capture and reason about the emotional content in the multimodal input.

By employing this linear projection and multimodal token representation, Emotion-LLaMA processes and integrates information from various modalities, leveraging the strengths of the underlying LLaMA model while incorporating essential emotional cues from audio and visual sources. Further details of the Emotion-LLaMA Model are provided in the code repository.

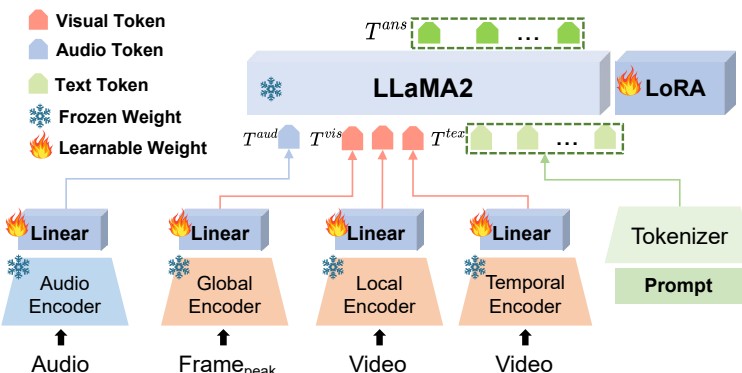

Figure 2: Architecture of Emotion-LLaMA, which integrates audio, visual, and text inputs for multimodal emotional recognition and reasoning.

## 3.3 Training of Emotion-LLaMA Model

We design a multi-task learning scheme to simultaneously supervise the model in learning emotional reasoning and recognition. The ground truth output and labels are converted and concatenated as standard text by a formatted template for autoregressive loss calculation [99]. Iterative random instruction sampling (see Table 16 and 17 for full list of instructions) for emotional reasoning and recognition tasks during training guides the model to develop a comprehensive understanding of emotions. Typically, Emotion-LLaMA is trained in a coarse-to-fine manner, consisting of the *Pre-training* and *Multimodal Instruction Tuning*:

**Stage 1: Pretraining.** Initially, the model is trained on 28,618 coarse-grained samples from the MERR dataset. Distinct tasks help the model grasp emotions from multiple perspectives. This phase involves simple descriptions or classifications, facilitating the rapid alignment of multimodal feature tokens ($\langle \mathrm{T}^{aud} \rangle$, $\langle \mathrm{T}^{vis}_{glo} \rangle$, $\langle \mathrm{T}^{vis}_{loc} \rangle$, and $\langle \mathrm{T}^{vis}_{temp} \rangle$) to the word embedding space [10, 97].

**Stage 2: Multimodal Instruction Tuning.** The pretrained Emotion-LLaMA model is then refined using fine-grained instructional datasets to enhance its capacity for emotion recognition and reasoning. This stage utilizes multimodal instruction tuning datasets, incorporating 4,487 fine-grained annotated descriptions for comprehensive reasoning from the MERR dataset. The tuning process is extended to diverse sources, including MER2023 [59] and DFEW [45], which feature precisely annotated emotional categories. This phase ensures that the model not only identifies emotions accurately but also understands the underlying context and reasoning behind each emotion. More details are in the code repository and the Appendix B.

# 4 Experiments

## 4.1 Experimental Setup

To verify the effectiveness of Emotion-LLaMA, we conducted extensive evaluations across four different datasets: MER2023 [59], MER2024 [60], DFEW [45], and EMER [62]. Notably, we utilized the MERR dataset for pre-training the model and then fine-tuned it on target datasets for evaluation.

**Emotion Recognition Evaluation**. We performed instruction tuning on the MER2023 and DFEW datasets, allowing the model to integrate the emotional knowledge acquired during pretraining. To test the generalizability of our model, we used three datasets: MER2023, MER2024, and DFEW. These datasets are multimodal emotion recognition datasets composed of movie and TV series clips, each annotated with various emotion categories. For fair comparisons, we evaluated Emotion-LLaMA on MER2023-SEMI and MER2024-NOISE using the F1 score. We also compared it with other MLLMs and state-of-the-art (SOTA) methods using unweighted average recall (UAR) and weighted average recall (WAR) on the DFEW dataset. Additionally, we used the average of accuracy and recall scores as evaluation metrics on the MER2024-OV dataset.

**Emotion Reasoning Evaluation**. The EMER dataset differs from traditional emotion datasets by including emotion trigger labels, such as facial micro-expressions, tone of speech, and video context information, in addition to emotion categories. To assess the emotional reasoning capabilities of different MLLMs on the EMER dataset, we employ ChatGPT to score their predictions, focusing on three key aspects: (1) the degree of overlap between emotion-related clues, (2) the degree of overlap between summarized emotional states, and (3) the completeness of the reasoning process across modalities. This multi-faceted evaluation provides a rigorous and in-depth assessment of the models' ability to understand and explain emotions in a multimodal context.

## 4.2 Implementation Details

For the global visual encoder, we employ the EVA model with full images sized at 448×448 pixels as input. For the local and temporal visual encoders, we first crop and align the faces within the images, then hierarchical sample 16 facial images as inputs for the MAE and VideoMAE models. The audio is handled by the HuBERT-Chinese large model. The extracted emotional features are transformed into a 4096-dimensional space via linear layers before being concatenated with text tokens.

During the tuning process, we froze the visual and audio backbones, focusing on training the linear projection layer. For the language model (LLM), we utilize LLaMA2-chat (7B) equipped with LoRA for parameter-efficient tuning. Following the Minigpt-v2 approach, we fine-tune the query and value projection matrices ($\mathcal{W}_q$ and $\mathcal{W}_v$) by setting $r = 64$ and $\alpha = 16$. Consequently, the trainable parameters of Emotion-LLaMA totaled only 34 million, representing a mere 0.495% of the overall parameter count. We train on 4*A100 GPUs for 300,000 steps, which takes around 20 hours. Detailed information can be found on the project homepage and in the code repository.

## 4.3 Comparison with State-of-the-Art Methods

To comprehensively evaluate the performance of Emotion-LLaMA, we compared it with several state-of-the-art (SOTA) methods across different datasets.

**Multimodal Emotion Reasoning Results.** We compared Emotion-LLaMA with contemporary MLLMs such as Video-LLaMA, Video-ChatGPT, PandaGPT, VideoChat, and Valley, presenting the results in Table 1. VideoChat demonstrates that aligning visual data directly with textual embedding space (VideoChat-Embed) significantly outperforms converting it into textual format (VideoChat-Text), supporting our method of mapping audio and visual features to textual embedding space. Notably, other MLLMs that accept audio inputs,

Table 1: Comparison of multimodal emotion reasoning results on the EMER dataset. Clue Overlap and Label Overlap scores range from 0 to 10.

| Models | Clue Overlap | Label Overlap |
|---|---|---|
| VideoChat-Text [54] | 6.42 | 3.94 |
| Video-LLaMA [97] | 6.64 | 4.89 |
| Video-ChatGPT [68] | 6.95 | 5.74 |
| PandaGPT [81] | 7.14 | 5.51 |
| VideoChat-Embed [54] | 7.15 | 5.65 |
| Valley [67] | 7.24 | 5.77 |
| Emotion-LLaMA (ours) | **7.83** | **6.25** |

like PandaGPT and Video-LLaMA, show no standout performance, suggesting inefficiencies in

extracting rich emotional content from audio. Emotion-LLaMA excels beyond these models across both Clue Overlap and Label Overlap evaluation metrics, highlighting our model's unparalleled ability to extract direct emotional features and engage in logical emotional reasoning. The scoring criteria and cases are presented in the Appendix C.1.

**Multimodal Emotion Recognition Results.** Table 2 presents the comparison results on the DFEW dataset. In the zero-shot scenario, Emotion-LLaMA demonstrates superior capabilities compared to all other MLLMs, showcasing its strong generalization ability. Notably, the majority of MLLMs scored zero in the disgust category, with GPT-4V achieving only 10.34%. This may be attributed to safety constraints on the term "disgust" within large language models, indicating a need for further exploration. Additionally, different MLLMs tend to favor predicting a specific emotion category, resulting in higher scores for those categories but lower recall scores overall. In contrast, Emotion-LLaMA maintains a more balanced prediction across all categories, ultimately achieving the highest WAR score of 59.37%. After fine-tuning, Emotion-LLaMA achieves the highest Unweighted Average Recall (UAR) and Weighted Average Recall (WAR) scores, further indicating its exceptional performance in emotion recognition tasks. These results highlight the effectiveness of our model in adapting to new datasets and accurately identifying emotions across various modalities. Overall, the results of Emotion-LLaMA's performance highlight the effectiveness of our approach in accurately recognizing emotions from multimodal data.

Table 2: Comparison of multimodal emotion recognition results on DFEW. The upper part shows zero-shot performance, while the lower part shows results after fine-tuning.

| Method | Hap | Sad | Neu | Ang | Sur | Dis | Fea | UAR | WAR |
|---|---|---|---|---|---|---|---|---|---|
| *Zero-Shot* | | | | | | | | | |
| Qwen-Audio [22] | 25.97 | 12.93 | 67.04 | 29.20 | 6.12 | 0.00 | 35.36 | 25.23 | 31.74 |
| LLaVA-NEXT [64] | 57.46 | **79.42** | 38.95 | 0.00 | 0.00 | 0.00 | 0.00 | 25.12 | 33.75 |
| MiniGPT-v2 [10] | **84.25** | 47.23 | 22.28 | 20.69 | 2.04 | 0.00 | 0.55 | 25.29 | 34.47 |
| Video-LLaVA(image) [63] | 37.09 | 27.18 | 26.97 | 58.85 | 12.97 | 0.00 | 3.31 | 20.78 | 31.10 |
| Video-LLaVA(video) [63] | 51.94 | 39.84 | 29.78 | 58.85 | 0.00 | 0.00 | 2.76 | 26.17 | 35.24 |
| Video-Llama [97] | 20.25 | 67.55 | **80.15** | 5.29 | 4.76 | 0.00 | 9.39 | 26.77 | 35.75 |
| GPT-4V [61] | 62.35 | 70.45 | 56.18 | 50.69 | 32.19 | 10.34 | **51.11** | **47.69** | 54.85 |
| Emotion-LLaMA (ours) | 71.98 | 76.25 | 61.99 | **71.95** | **33.67** | 0.00 | 3.31 | 45.59 | **59.37** |
| *Fine-tuning* | | | | | | | | | |
| EC-STFl [45] | 79.18 | 49.05 | 57.85 | 60.98 | 46.15 | 2.76 | 21.51 | 45.35 | 56.51 |
| Former-DFER [102] | 84.05 | 62.57 | 67.52 | 70.03 | 56.43 | 3.45 | 31.78 | 53.69 | 65.70 |
| IAL [52] | 87.95 | 67.21 | 70.10 | 76.06 | 62.22 | 0.00 | 26.44 | 55.71 | 69.24 |
| MAE-DFER [82] | 92.92 | 77.46 | 74.56 | 76.94 | 60.99 | **18.62** | 42.35 | 63.41 | 74.43 |
| VideoMAE [84] | 93.09 | 78.78 | 71.75 | 78.74 | 63.44 | 17.93 | 41.46 | 63.60 | 74.60 |
| S2D [12] | **93.62** | **80.25** | **77.14** | 81.09 | 64.53 | 1.38 | 34.71 | 61.82 | 76.03 |
| Emotion-LLaMA (ours) | 93.05 | 79.42 | 72.47 | **84.14** | **72.79** | 3.45 | **44.20** | **64.21** | **77.06** |

## 4.4 Multimodal Emotion Recognition Challenge

To further validate the effectiveness of our proposed Emotion-LLaMA model, we conducted experiments on the MER2023 and MER2024 Challenge, comparing it with previous state-of-the-art methods.The results, presented in Table 3, demonstrate that our model, which maps audio and visual features to the textual space, achieves the highest F1 score across various modalities. This approach significantly enhances the context of the textual modality by providing a more comprehensive understanding of the information, thereby outperforming other models. By integrating audio, visual, and textual data, Emotion-LLaMA can better capture the nuances of emotional expression, leading to more accurate and reliable emotion recognition.

The MER2024 Challenge introduced a new Open-Vocabulary Multimodal Emotion Recognition (MER-OV) task. Unlike traditional tasks, MER-OV focuses on recognizing any number of labels across diverse categories, aiming for a more nuanced and precise description of emotional states. As shown in Table 4, Emotion-LLaMA outperforms other mainstream multimodal models, yielding an 8.52% improvement in average accuracy and recall compared to GPT-4V, and achieving the highest zero-shot score among all participating large multimodal models. These results showcase the robustness and versatility of our approach in handling complex multimodal data for emotion recognition tasks, making it a promising solution for real-world applications.

Table 3: Comparison of multimodal emotion recognition results on MER2023. The table shows the performance of different models across various modalities, with the highest F1 scores achieved by our proposed method.

| Method | Modality | F1 Score |
|---|---|---|
| wav2vec 2.0 [4] | A | 0.4028 |
| VGGish [38] | A | 0.5481 |
| HuBERT [39] | A | 0.8511 |
| ResNet [37] | V | 0.4132 |
| MAE [36] | V | 0.5547 |
| VideoMAE [84] | V | 0.6068 |
| RoBERTa [66] | T | 0.4061 |
| BERT [25] | T | 0.4360 |
| MacBERT [24] | T | 0.4632 |
| MER2023-Baseline [59] | A, V | 0.8675 |
| MER2023-Baseline [59] | A, V, T | 0.8640 |
| Transformer [8] | A, V, T | 0.8853 |
| FBP [13] | A, V, T | 0.8855 |
| VAT [26] | A, V | 0.8911 |
| Emotion-LLaMA | A, V | 0.8905 |
| Emotion-LLaMA | A, V, T | **0.9036** |

Table 4: Performance (%) of Multimodal Large Language Models on MER2024 Challenge track 3: MER-OV. The "avg" column represents the average of "Accuracy$_S$" and "Recall$_S$"

| Model | Accuracy$_S$ | Recall$_S$ | Avg |
|---|---|---|---|
| Empty | 0.00 | 0.00 | 0.00 |
| Random | 13.42 | 24.85 | 19.13 |
| Ground Truth | 93.37 | 52.51 | 72.94 |
| Valley [67] | 20.16 | 13.26 | 16.71 |
| Otter [51] | 29.64 | 23.04 | 26.34 |
| PandaGPT [81] | 35.75 | 31.57 | 33.66 |
| Video-LLaMA [97] | 31.08 | 32.26 | 31.67 |
| VideoChat [54] | 43.17 | 44.92 | 44.05 |
| VideoChat2 [55] | 46.91 | 34.78 | 40.85 |
| Video-ChatGPT [68] | 46.20 | 39.33 | 42.77 |
| SALMONN [83] | 42.20 | 44.75 | 43.47 |
| Qwen-Audio [22] | 55.12 | 32.91 | 44.02 |
| mPLUG-Owl [94] | 44.80 | 46.54 | 45.67 |
| AffectGPT [62] | 66.14 | 46.56 | 56.35 |
| GPT-4V [71] | 56.19 | 58.97 | 57.58 |
| Emotion-LLaMA | **69.61** | **62.59** | **66.10** |

Table 5: An example of multimodal emotion reasoning comparing Emotion-LLaMA with other MLLMs. Incorrect reasoning is marked in red, correct reasoning in blue, and hallucinations in gray.

**An Example of Multimodal Emotion Reasoning**

| | |
|---|---|
| Subtitle (En) | Are you really stupid or just pretending? Do I care about spending this money? It's just a joke. |
| Ground Truth | In the video, although she had a smile on her face, she questioned the other person, "Are you really stupid or just pretending?" expressing her dissatisfaction and anger towards the other person's act of pretending to be foolish. Therefore, it can be inferred that his emotional state is angry. |
| PandaGPT | The emotional state of this person appears to be positive and relaxed. They are smiling and laughing, indicating that they find the situation amusing or enjoyable. The woman's reaction to the man's joke suggests she is not offended or angry but finds it funny or amusing. This means that the interaction between the two individuals is friendly, not tense or uncomfortable. |
| VideoChat-Embed | Based on the video, the lady appears to be quite upset and angry. She discusses her financial losses and how her boyfriend seems indifferent to her feelings. She expresses a desire to spend money. The lady's facial expressions and gestures, such as frowning and clutching her purse tightly... |
| Valley | From the video, we can infer that the woman might be in a neutral or mixed emotional state. In some scenes, she is smiling or making amusing faces, suggesting she feels happy or amused. . . |
| Emotion-LLaMA | In the video, the woman's facial expressions, including a furrowed brow, lifted eyebrows, and a slightly lowered lip, suggest a mix of emotions. Her tone and intonation convey a sense of frustration, and her words reveal that she is questioning whether the other person is silly, indicating a sense of annoyance. Therefore, it is inferred that her emotional state is anger. |

## 4.5 Qualitative Analysis of Emotion Reasoning

To illustrate the qualitative performance of Emotion-LLaMA, we present a detailed comparison of emotion reasoning results across different models. Table 5 displays the emotion reasoning results of the four highest-scoring models. The video shows a person smiling while questioning another individual, an expression of dissatisfaction that suggests an angry emotional state. Accurate emotion reasoning for this sample necessitates integrating information from multiple modalities. PandaGPT and Valley captured the correct visual features but failed to incorporate information from other modalities, incorrectly classifying the emotion as happy. In contrast, VideoChat-Embed eventually reached the correct inference, but its reasoning was compromised by hallucinations. Emotion-LLaMA went a step further by recognizing the tone of the person and combining subtle facial expressions with multimodal information for accurate emotion reasoning. This example demonstrates the superiority of our model in understanding and integrating emotional cues from various modalities, resulting in more precise and contextually relevant emotion recognition.

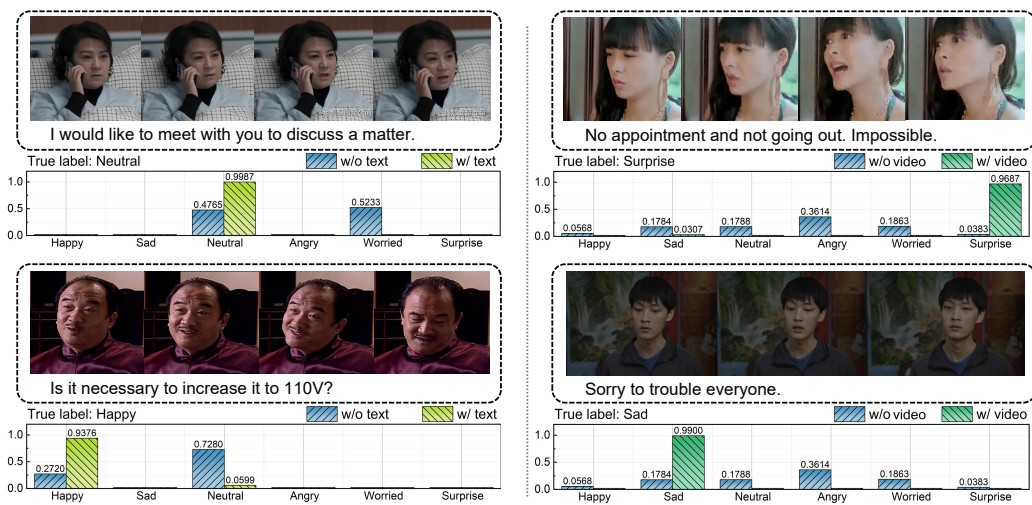

Figure 3: Visualization of the output probability distribution for multimodal emotion recognition by different models. Each sample is represented by two bar graphs: the left graph displays the results from other models, and the right graph shows the results from Emotion-LLaMA.

Figure 3 displays the recognition results of Emotion-LLaMA in comparison to other models. The left samples show that even when characters exhibit subtle emotional fluctuations lacking distinct emotional features, our Emotion-LLaMA model accurately discerns the true intentions behind the text. The right samples demonstrate that, unlike other language models, Emotion-LLaMA can extract multimodal emotional features to enhance the text, thereby accurately understanding the true emotions conveyed by the text. These qualitative results further illustrate the effectiveness of our model in capturing and interpreting emotional nuances across different modalities, leading to a more comprehensive and accurate understanding of human emotions.

## 4.6 Ablation Evaluation

We conducted a series of ablation experiments to explore the effectiveness of each component of the proposed Emotion-LLaMA. More ablation experiments, which examine factors affecting instruction-tuning performance, including data quantity and quality as well as hyperparameters, are presented in Appendix C.

**Investigation of Encoders**. We explore various combinations of encoders. As shown in Table 6, the combinations listed in the upper part are fused by the part of the modalities, while the rest capture all the modalities. The best combination of the audio and multiview visual encoders is the HuBERT+MAE+VideoMAE+EVA, which obtained a 0.891 F1 score. Two observations are worth noting: 1) multiple modalities, including audio, static, and dynamic vision, are compensated for in the emotion capturing process; 2) the spatial, spatial-context, and temporal information are fully considered by the multiview visual module, which exhibits a significant improvement over single-view approaches. These findings highlight the importance of incorporating diverse modalities and considering multiple aspects of visual information for accurate emotion recognition.

Table 6: Ablation Study Results for Different Encoders.

| Audio Encoder | Visual Encoder | F1 Score |
|---|---|---|
| Wav2Vec | - | 0.4893 |
| VGGish | - | 0.5944 |
| whisper | - | 0.5324 |
| HuBERT | - | 0.8394 |
| - | MAE | 0.6366 |
| - | VideoMAE | 0.6762 |
| - | EVA | 0.6635 |
| - | MAE, VideoMAE, EVA | 0.7122 |
| HuBERT | MAE | 0.8800 |
| HuBERT | VideoMAE | 0.8805 |
| HuBERT | EVA | 0.8757 |
| HuBERT | MAE, VideoMAE | 0.8880 |
| HuBERT | MAE, EVA | 0.8896 |
| HuBERT | VideoMAE, EVA | 0.8802 |
| HuBERT | MAE, VideoMAE, EVA | **0.8910** |

In Table 7, we present the impact of different instruction data on the instruction-tuning of Emotion-LLaMA. 'Raw' refers to the direct concatenation of visual and audio descriptions as instructions for training Emotion-LLaMA, which yielded the poorest performance. When trained using the coarse-grained set from the MERR dataset, Emotion-LLaMA achieved scores of 7.41 and 5.56 for clue and label overlap, respectively. This marks an improvement of 1.87 and 1.25 over the 'Raw' approach, demonstrating that coarse-grained annotations generated by the LLaMA-3 model effectively integrate emotional cues to capture genuine emotional expressions. Notably, further instruction-tuning using the fine-grained set from the MERR dataset resulted in additional gains of 0.42 and 0.69 in clue and label overlap, respectively, indicating that fine-grained annotations offer higher quality data and further enhance the performance of instruction-tuning.

To better understand the effect of the sample selection strategy, we compared our strategy against traditional semi-supervised approaches, as shown in Table 8. Due to the limited size of the MER2023 training dataset, which contains only 3,373 samples, pre-training on this dataset can lead to difficulties in fitting models with transformer structures, resulting in a low F1 score of 0.7977. We then compared traditional semi-supervised approaches, which involve assigning pseudo-labels to all unlabeled samples or selectively giving pseudo-labels to those samples that exhibit high softmax scores during inference. There are 73,148 and 36,490 samples selected by these two strategies, respectively. During the pre-training phase, a substantial increase in data volume can significantly enhance performance, even introducing considerable noise. Ultimately, the model tuning on our automatically annotated MERR dataset achieves the best model performance. This demonstrates the effectiveness and robustness of the proposed Emotion-LLaMA in leveraging large-scale, diverse datasets for improved emotion recognition and reasoning.

Table 7: Ablation Study Results for Different Stages of the MERR Dataset Instructions.

| Stage | Clue Overlap | Label Overlap |
|---|---|---|
| Raw | 5.54 | 4.31 |
| Coarse-grained | 7.41 | 5.56 |
| Fine-grained | 7.83 | 6.25 |

Table 8: Ablation Study Results for Different Training Strategies.

| Strategy | F1 Score |
|---|---|
| Standard Training | 0.7917 |
| High-Confidence Training | 0.8831 |
| Pseudo-Label Training | 0.8950 |
| Instruction Tuning | **0.9036** |

## 5 Ethics and Conclusion

**Ethics.** All datasets used in this study are governed by signed usage agreements, strictly restricting their use to academic research. The MERR dataset is derived from MER2023 and includes over 70,000 unannotated samples from diverse movies and TV series. We have obtained the necessary End User License Agreements (EULA) and explicit permissions from the original data providers. The open-source MERR dataset contains only emotion description JSON files, intentionally excluding source videos. Researchers must apply directly to the original providers and fully comply with the EULA to access the dataset. We ensure that the MERR dataset exclusively comprises multimodal emotion descriptions without any discriminatory or biased content.

**Conclusion.** We introduced *Emotion-LLaMA*, a novel multimodal large language model designed to accurately recognize and interpret human emotions in real-world scenarios. Utilizing robust open-source tools, we automated the selection and annotation of the Multimodal Emotion Recognition and Reasoning (MERR) dataset for pre-training purposes. Instruction-tuning on comprehensive datasets such as MER2023 and DFEW enabled us to achieve state-of-the-art performance metrics. Comparative analyses with other advanced multimodal large language models (MLLMs) demonstrated Emotion-LLaMA's superior generalization capabilities in emotion recognition and reasoning tasks.

**Acknowledgments.** This work was partially supported by the National Natural Science Foundation of China (Grant No. 62176165), Stable Support Projects for Shenzhen Higher Education Institutions (Grant No. 20220718110918001), and the Natural Science Foundation of Top Talent at SZTU (Grant No. GDRC202131). Additional support was provided by NSF CISE (Grant No. 1937998), the Air Force Research Laboratory (Grant No. FA8750-19-2-0200), U.S. Department of Commerce (Grant No. 60NANB17D156), IARPA (Grant No. D17PC00340), and DARPA (Grant No. HR00111990063). The research findings and conclusions presented in this paper are solely those of the authors and do not necessarily represent the views of the funding agencies.

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

# A MERR Dataset Details

## A.1 Emotion Categories and Annotations

The Multimodal Emotion Recognition and Reasoning (MERR) dataset covers a diverse range of emotion categories, including some that are often overlooked or challenging to distinguish, as shown in Figure 4 and Figure 5. The dataset includes nine emotion categories: neutral, happy, angry, worried, surprise, sad, fear, doubt, and contempt. While the first seven categories are commonly addressed in most emotion datasets, MERR stands out by also focusing on doubt and contempt. These two categories are often underrepresented due to the difficulty in collecting sufficient samples and the potential for confusion with other emotions. Doubt, for example, can easily be mistaken for worry, as both emotions involve a sense of uncertainty and concern. However, there are subtle differences in facial expressions and contextual cues that can help distinguish between the two. Doubt often involves a more questioning or skeptical facial expression, with raised eyebrows and a slight frown, whereas worry tends to have a more anxious or apprehensive appearance, with furrowed brows and a downturned mouth. Contempt, on the other hand, is frequently misclassified as happiness due to the presence of a smile. However, the smile associated with contempt is often a scornful or dismissive one, accompanied by a slight sneer or a raised upper lip. The context and manner in which the smile is displayed can help differentiate between genuine happiness and contemptuous expression.

To accurately categorize these challenging emotions, the MERR dataset relies on rich multimodal descriptions that provide a comprehensive understanding of the emotional state and its context. These descriptions go beyond simple categorical labels and offer detailed insights into the facial expressions, body language, vocal cues, and situational factors that contribute to the emotional interpretation. Table 12 presents a template for the multimodal descriptions used in MERR, showcasing the different components that are captured for each sample. The descriptions include a visual expression component that focuses on the specific facial movements and action units associated with the emotion, a visual objective component that describes the overall scene and context, an audio tone component that captures the vocal cues and intonation, and a textual component that provides the transcribed speech or dialogue. To further illustrate the value of these detailed annotations, Tables 13 and 14 provide specific examples of annotated samples for doubt and contempt, respectively.

## A.2 Data Filtering and Pseudo-Labeling

To create a high-quality dataset for multimodal emotion recognition and reasoning, we employed a data filtering and pseudo-labeling process. This process aimed to identify video segments with strong emotional expressions and assign them initial emotion labels based on facial cues.

First, we used OpenFace[6] to extract faces from the video segments. OpenFace is a state-of-the-art tool for facial behavior analysis that can detect and track facial landmarks, head pose, eye gaze, and facial action units (AUs). AUs are a widely used system for describing facial muscle movements, with each AU corresponding to a specific muscle or group of muscles. After extracting the faces, we aligned them to a canonical pose to facilitate accurate AU detection. OpenFace then analyzed the aligned faces to identify the presence and intensity of various AUs throughout each video segment. Next, we utilized the detected AUs to assign pseudo-labels to the video segments. As shown in Table 10, certain combinations of AUs are strongly correlated with specific emotions. For example, the combination of AU05 (upper lid raiser) and AU26 (jaw drop) is often associated with the emotion of surprise. Similarly, the presence of AU04 (brow lowerer) and AU15 (lip corner depressor) is indicative of sadness. By leveraging these known AU combinations, we created a rule-based system to assign pseudo-labels to the video segments. If a segment exhibited a specific combination of AUs with sufficient intensity, it was assigned the corresponding emotion label. This process allowed us to identify samples that displayed strong emotional expression characteristics based on facial cues alone. Through this data filtering and pseudo-labeling process, we selected a total of 28,618 samples from the initial pool of video segments. These samples were assigned pseudo-labels corresponding to the nine emotion categories in the MERR dataset: neutral, happy, angry, worried, surprise, sad, fear, doubt, and contempt. It is important to note that while the pseudo-labels provided a valuable starting point for annotation, they were not relied upon as ground truth. In subsequent stages of the dataset construction process, human annotators reviewed and refined these labels, taking into account additional context from the visual, audio, and textual modalities.

---

[6]https://github.com/TadasBaltrusaitis/OpenFace

### A.3 Instruction Collection and Multimodal Annotation

To provide rich, multimodal annotations for the MERR dataset, we collected instructions and descriptions from various sources, focusing on four key aspects: visual expression, visual context, audio tone, and multimodal integration.

**Visual Expression Description.** In videos, natural actions such as blinking and speaking can lead to different combinations of Action Units (AUs) being extracted from various frames. To accurately represent the current emotion, it is crucial to determine the most relevant AUs. As illustrated in Figure 6, our approach involves analyzing the amplitude values of the AUs to identify the "emotional peak frame", which captures the most intense emotional expression.

The specific steps for identifying the emotional peak frame are as follows:

1. Identify the AUs present in all frames of the video segment.
2. Sum the amplitude values of these AUs for each frame.
3. Determine whether the combinations of these AUs match the pseudo-label.
4. Select the frame with the highest total amplitude as the emotional peak frame.

Once the emotional peak frame is determined, its corresponding AUs are mapped to visual expression descriptions using the guidelines provided in Table 11. These descriptions provide a detailed account of the facial movements and expressions associated with the emotion displayed in the peak frame.

**Visual Objective Description.** To provide a comprehensive understanding of the emotional context in each video, we utilize the MiniGPT-v2 [10] model to generate descriptions of the visual scene. By inputting the complete emotional peak frame, which captures the most intense emotional expression, MiniGPT-v2 can analyze and describe various aspects of the video, such as the environment, character actions, and object interactions. These visual objective descriptions offer valuable insights into the situational context surrounding the emotional expression. For example, if a character is shown in a dimly lit room with a concerned facial expression, the model might generate a description like "The scene takes place in a dark, shadowy room. The character appears to be sitting alone, with a worried look on their face, fidgeting with their hands." This description provides additional information about the setting and the character's body language, which can help in interpreting their emotional state. Moreover, the model can also identify and describe relevant objects or elements in the scene that may contribute to the emotional context. For instance, if a character is holding a letter and appears upset, the model might mention the presence of the letter in its description, suggesting that the content of the letter could be related to the character's emotional response.

**Audio Tone Description.** In addition to visual cues, audio plays a crucial role in conveying emotional information. The tone, intonation, and prosodic features of a speaker's voice can provide valuable insights into their emotional state, often revealing subtle nuances that may not be apparent from visual cues alone. To capture these audio cues, we employ the Qwen-Audio [22] model, which is specifically designed to analyze and describe the emotional content of speech. By processing the audio track of each video segment, it can generate detailed descriptions of the speaker's tone and intonation. For example, if a character is speaking with a trembling voice and a high pitch, the model might generate a description like "The speaker's voice is shaky and high-pitched, indicating a sense of fear or anxiety. There are noticeable pauses and hesitations in their speech, suggesting uncertainty or distress." This description captures the emotional nuances conveyed through the speaker's vocal delivery, providing additional context for understanding their emotional state. Moreover, Qwen-Audio can also identify and describe other relevant audio cues, such as sighs, laughter, or changes in speaking rate, which can further contribute to the emotional interpretation. For instance, if a character is speaking rapidly and laughing, the model might generate a description like "The person in the video speaks with a cheerful tone."

**Multimodal Description.** To generate initial coarse-grained emotional descriptions, we concatenate the information obtained from all modalities (visual expression, visual context, audio tone, and textual content). These surface-level descriptions for the 28,618 pseudo-labeled samples are used in the first stage of pre-training to help the model align emotional features with the textual semantic space. We input all the collected emotional clues into the LLaMA-3 model for further refinement. LLaMA-3 sifts through the clues, identifies the most relevant ones, and combines them to generate a comprehensive emotional description. This process helps to filter out any erroneous or contradictory

descriptions that may have been present in the initial set of emotional clues. Additionally, we remove duplicate or overabundant samples to ensure a balanced and diverse dataset. Through these refinement processes, the final MERR dataset contains 4,487 samples, each accompanied by a detailed multimodal description. Table 12 presents an example of the annotation format used for each sample in the dataset. Finally, four domain experts manually review the refined descriptions and use a voting process to select fine-grained samples, assessing whether the video descriptions are reasonable and if the emotional reasoning is accurate. By collecting and integrating instructions and descriptions from multiple modalities, we have created a rich and informative dataset that captures the complexities of emotional expressions in real-world scenarios. The multimodal annotations in MERR enable models to learn more comprehensive and nuanced representations of emotions, leading to improved performance on emotion recognition and reasoning tasks.

## A.4 Data Statistics and Comparisons

**Video Duration Distribution.** Figure 7 presents the distribution of video durations in the MERR dataset. The majority of the samples have a length between 2 and 4 seconds, which aligns with the typical duration of short, emotionally expressive video clips. This duration range strikes a balance between capturing sufficient context for emotion recognition and maintaining a manageable data size for processing and annotation. Shorter clips may lack the necessary context to fully understand the emotional state, while longer clips can be more challenging to annotate and may contain multiple or changing emotions. The concentration of samples in the 2-4 second range also reflects the natural temporal dynamics of emotional expressions. Most emotions are conveyed through relatively brief, intense bursts of facial movements, vocalizations, and body language. By focusing on this duration range, the MERR dataset captures the core expressive moments while minimizing the inclusion of neutral or ambiguous segments. Furthermore, the consistent duration range across the dataset facilitates the development of emotion recognition models that can operate on fixed-length input sequences. This consistency simplifies the data preprocessing and model architecture design, as the models can be optimized for the specific temporal scale of the emotional expressions.

**Comparison with Previous Datasets.** To highlight the unique features and contributions of the MERR dataset, we compare it with several related datasets in Table 15. The MER2023 [59] and DFEW [45] datasets primarily focus on discrete emotion category labels, providing a classification-oriented perspective on emotion recognition. While these datasets are valuable for developing models that can predict specific emotion categories, they lack the detailed descriptions and contextual information necessary for deeper emotion understanding. On the other hand, datasets like EmoSet [93] and EmoVIT [92] offer visual descriptions of emotional expressions, capturing the facial cues and body language associated with different emotions. However, these datasets do not include information from other modalities, such as audio or text, which can provide crucial insights into the emotional context and help disambiguate complex or subtle expressions. The EMER [62] dataset stands out as one of the few existing datasets that contain multimodal emotion descriptions, incorporating information from visual, audio, and textual modalities. However, due to the high cost and effort involved in manual annotation, EMER is limited to only 100 samples, which may not be sufficient for training robust and generalizable emotion recognition models.

In contrast, the MERR dataset offers a comprehensive and large-scale resource for multimodal emotion recognition and reasoning. With 28,618 coarse-grained and 4,487 fine-grained annotated samples, MERR provides a diverse range of emotional expressions across nine categories, including challenging ones like doubt and contempt. The extensive multimodal descriptions in MERR, encompassing visual expressions, visual context, audio tone, and textual information, enable a holistic understanding of emotional states and their situational context. Moreover, the MERR dataset's inclusion of detailed emotion reasoning annotations sets it apart from other datasets. By capturing the thought process and rationale behind the emotion labels, these annotations facilitate the development of models that can not only recognize emotions but also explain and justify their predictions. This level of interpretability is crucial for building trust and transparency in human-computer interaction scenarios. As such, MERR has the potential to advance the field of affective computing and contribute to the development of more intelligent and empathetic human-computer interaction systems.

**Assessing the Quality of Dataset Instructions.** We conducted a human evaluation of the annotation process by randomly selecting 20 video samples from each of the nine emotion categories. We then randomly shuffled the fine-grained annotations. Five volunteers evaluated the consistency and relevance of the video descriptions by scoring them. Each volunteer rated a total of 180 descriptions on a scale from 0 to 5. The evaluation criteria included:

1. Accuracy of the visual modality description.
2. Accuracy of the audio modality description.
3. Accuracy of the textual modality description.
4. Correctness of the reasoning process.
5. Correctness of the reasoning result.

As shown in the Table 9, the average score for the human evaluation of the MERR dataset is 4.258, indicating that the annotations are of high quality and align with real-world logic. Secondly, the "Neutral" category received the lowest score among all categories, suggesting that when a character's emotion is neutral, the facial and audio cues are weaker, making automatic annotation more challenging. We will address these findings and discuss related limitations in future work. Details of the human evaluation, including the code and assessment results, can be accessed through the code repository.

Table 9: The scores from manual evaluation of the MERR datasets for fine-grained annotation. The scores range from 0 to 5, with the Mean representing the average score across all categories.

| Volunteer | Angry | Happy | Surprise | Fear | Sad | Worry | Neutral | Doubt | Contempt | Mean |
|-----------|-------|-------|----------|------|------|-------|---------|-------|----------|------|
| Human-1 | 4.23 | 4.33 | 4.31 | 4.19 | 4.27 | 4.33 | 4.04 | 4.55 | 4.79 | 4.34 |
| Human-2 | 4.54 | 4.78 | 4.88 | 4.81 | 4.60 | 4.62 | 4.65 | 4.55 | 4.68 | 4.67 |
| Human-3 | 3.92 | 4.11 | 4.25 | 4.50 | 4.00 | 4.24 | 3.78 | 4.27 | 4.05 | 4.12 |
| Human-4 | 3.69 | 3.94 | 3.62 | 3.69 | 3.93 | 4.19 | 3.39 | 4.00 | 4.53 | 3.90 |
| Human-5 | 3.92 | 4.72 | 4.25 | 4.06 | 4.00 | 4.19 | 4.17 | 4.32 | 4.58 | 4.26 |

Table 10: Correspondence between facial expression labels and Action Units (AUs). Each expression label is associated with a unique combination of AUs, allowing for accurate mapping between facial movements and emotional categories.

- "happy": [ "AU06", "AU12", "AU14"]
- "angry": [ "AU04", "AU05", "AU07", "AU23", "AU10", "AU17"]
- "worried": [ "AU28", "AU20"]
- "surprise": [ "AU01", "AU02", "AU05", "AU26"]
- "sad": [ "AU04", "AU01", "AU14", "AU15"]
- "fear": [ "AU01", "AU02", "AU04", "AU05", "AU07", "AU20", "AU26"]
- "doubt": [ "AU25"]
- "contempt": [ "AU12", "AU10", "AU15", "AU17"]

Table 11: Mapping of Action Units (AUs) to their corresponding textual descriptions. Each AU represents a specific facial muscle movement, and the textual descriptions provide a human-interpretable explanation of the visual cues associated with each AU.

- "AU01": ["Inner brow raiser", "Frown", "Eyebrow raised", "Head lifting wrinkles", "Lift eyebrows"]
- "AU02": ["Outer brow raiser", "Outer brow lift", "Elevate outer brow", "Outer brow arch"]
- "AU04": ["Brow Lowerer", "Frowns furrowed", "Lower eyebrows", "A look of disapprroval"]
- "AU05": ["Upper Lid Raiser", "Pupil enlargement", "Eyes widened", "Lift upper eyelids", "Raise upper eyelids"]
- "AU06": ["Cheek Raiser", "smile, Pleasure", "Slight decrease in eyebrows", "Eyes narrowing", "Slightly lower eyebrows"]
- "AU07": ["Lid Tightener", "Facial tightness", "Tightening of eyelids"]
- "AU09": ["Nose Wrinkler", "Wrinkle the nose", "Curl the nose", "Make a face", "Pucker the nose"]
- "AU10": ["Upper Lip Raiser", "Curl the lips upwards", "Upper lip lift", "Lips apart showing teeth"]
- "AU12": ["Lip Corner Puller", "Toothy smile", "Grinning", "Big smile", "Show teeth"]
- "AU14": ["Dimpler", "Cheek dimple", "Indentation when smiling", "Hollow on the face when smiling"]
- "AU15": ["Lip Corner Depressor", "Downturned corners of the mouth", "Downward mouth curvature", "Lower Lip Depressor"]
- "AU17": ["Chin Raiser", "Lift the chin", "Chin held high", "Lips arching", "Lips forming an upward curve"]
- "AU20": ["Lip stretcher", "Tense lips stretched", "Anxiously stretched lips", "Nasal flaring", "Nostrils enlarge"]
- "AU23": ["Lip Tightener", "Tighten the lips,' 'Purse the lips", "Press the lips together"]
- "AU25": ["Lips part", "Open the lips", "Slightly puzzled", "lips slightly parted"]
- "AU26": ["Jaw Drop", "Mouth Stretch", "Open mouth wide", "Wide-mouthed", "Lips elongated"]
- "AU28": ["Lip Suck", "Purse lips", "Pucker lips", "Draw in lips", "Bring lips together"]

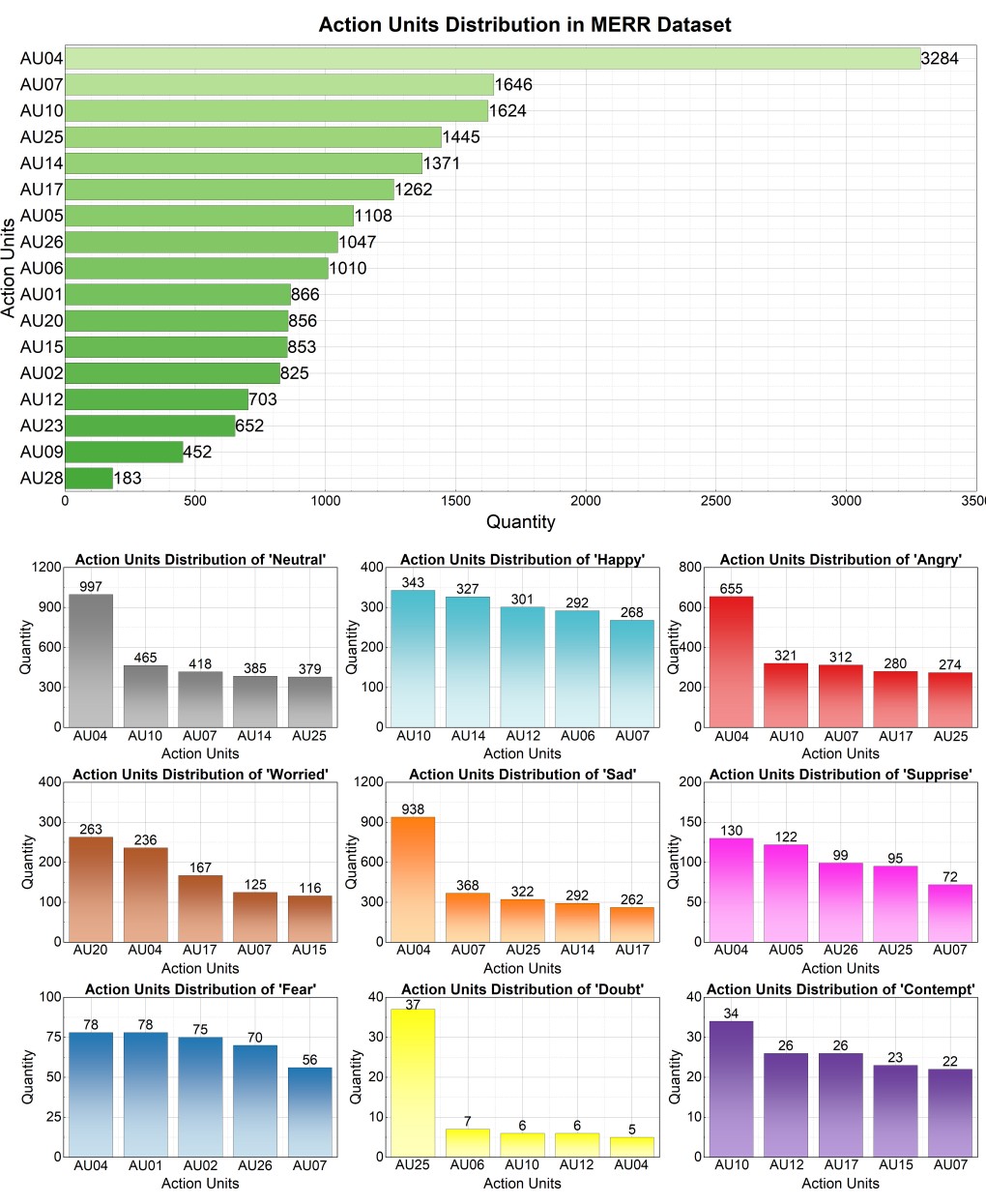

Figure 4: Distribution and analysis of Action Units (AUs) in the MERR dataset. The top part displays the frequency of different AUs across all samples, highlighting the most prevalent facial movements. The bottom part presents a statistical breakdown of the top five most frequent AUs for each of the nine facial expression categories, providing insights into the facial patterns associated with each emotion.

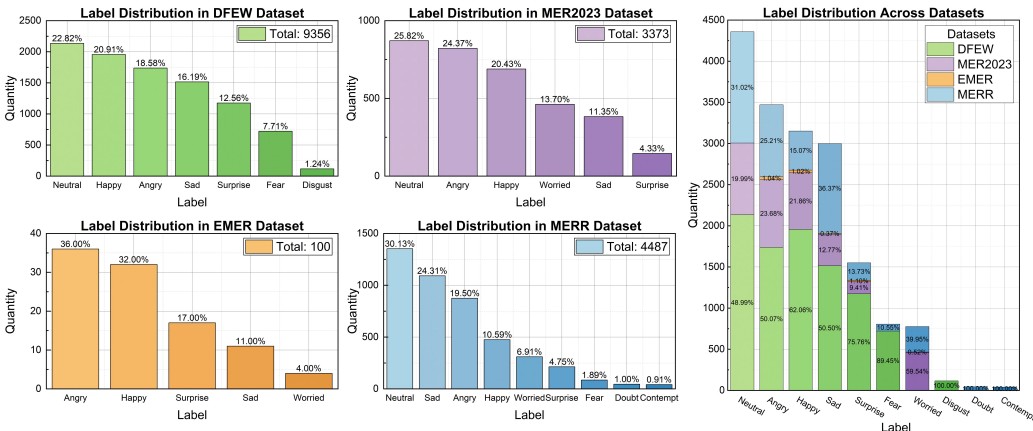

Figure 5: Comparative analysis of label distributions across the DFEW, MER2023, EMER, and MERR datasets. The left side illustrates the individual label distributions for each dataset, while the right side presents a side-by-side comparison of the label distributions, highlighting the similarities and differences in emotional category representation among the datasets.

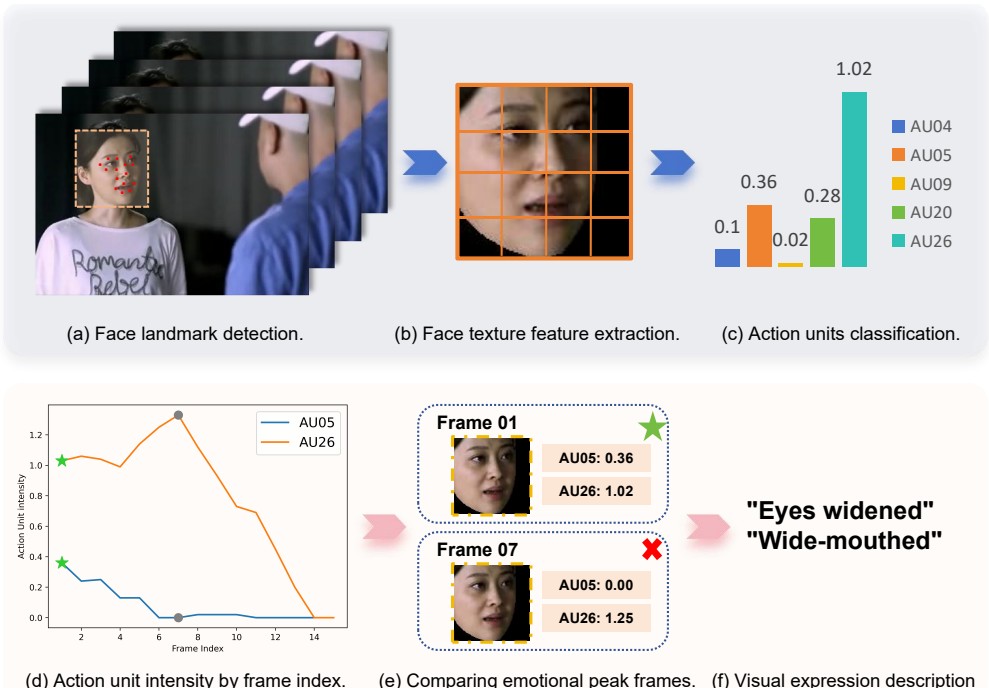

Figure 6: Overview of the video expression annotation process using Action Units (AUs). The top section illustrates the sampling of frames from a video, extraction of aligned faces, and measurement of AU intensities in each frame. The bottom section depicts the identification of the emotional peak frame based on the highest total AU intensity. The AUs of the peak frame are then translated into corresponding facial expression descriptions, providing a detailed characterization of the emotional expression at the most salient moment in the video.

Table 12: Example of an instruction-following data sample and its corresponding instruction template. The template provides a structured format for presenting the multimodal data, including visual, audio, and textual information, along with the target instruction and output. This structured representation facilitates the training and evaluation of the Emotion-LLaMA model on instruction-following tasks.

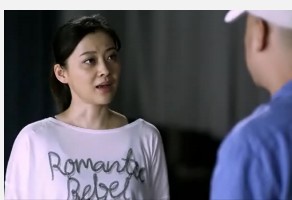

**Lexical subtitle:**
 "Oh my!"
**Audio tone description:**
 "The woman in the video speaks with an excited voice."
*Visual expression description:*
 "Eyes widened, Wide-mouthed."
*Visual objective description:*
 "The woman in the video is talking to a man, possibly discussing something important or sharing her thoughts and feelings."

---

**Coarse-grained annotation synthesis:**
Template: "This person in the video is {Visual objective description}. The person's expression and actions include {Visual expression description}, and the person's tone is {Auditory tone description}. Saying: {Lexical subtitle}"

Example: "The woman in the video is talking to a man, possibly discussing something important or sharing her thoughts and feelings. The woman's expression and actions include eyes widened and mouth wide open, and the woman in the video speaks with an excited voice. Saying: Oh my!"

---

**Fine-grained description generation by LLaMA-3:**
System: You are an emotion analysis expert. Please infer emotion label based on the given the emotional features.

Question: The woman in the video is talking to a man, possibly discussing something important or sharing her thoughts and feelings. The person's expression and action include eyes widened, wide-mouthed, and speaking with a happy voice, saying: "Oh my." Please sort out the correct emotional clues and infer why the person in the video feels surprise.

Answer: In the video, a woman is conversing with a man. Her facial expressions, with eyes widened and mouth wide open, clearly show surprise. She amplifies this visual cue with an excited tone as she exclaims, "Oh my." This combination of voice and expression indicates that she is experiencing surprise, likely in response to unexpected news shared during the conversation.

Table 13: Example of the emotion category "doubt". The table presents a multimodal data sample associated with the "doubt" category, including the visual expression description, audio tone description, and textual content. This example highlights the subtle and complex nature of the "doubt" emotion and the importance of considering multiple modalities for accurate recognition and reasoning.

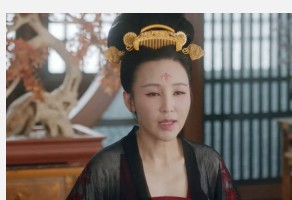

**Lexical subtitle:**
 "That must not be his real name."
**Audio tone description:**
 "The woman's tone sounds natural."
*Visual expression description:*
 "Lips slightly parted."
*Visual objective description:*
 "The person in the video is a woman wearing traditional Chinese clothing and a gold headpiece."
**Multimodal description:**
 "In the video, the woman's lips are slightly parted. Her tone is natural, but her words express skepticism as she says, 'That must not be his real name.' Her questioning of the authenticity of the name suggests that she is experiencing doubt."

Table 14: Example of the emotion category "contempt". The table showcases a multimodal data sample corresponding to the "contempt" emotion, encompassing the visual expression description, audio tone description, and textual content. This example illustrates the nuanced and challenging characteristics of the "contempt" category, emphasizing the need for a comprehensive multimodal approach to capture its subtleties.

**Lexical subtitle:**
    "Isn't he who he is?"
**Audio tone description:**
    "The person in the video speaks in a normal tone."
**Visual expression description:**
    "Downturned corners of the mouth, lips apart showing teeth, lip Corner Puller, chin held high."
**Visual objective description:**
    "The person in the video is a man with long black hair and is wearing a white shirt and a brown vest."
**Multimodal description:**
    "In the video, the person's facial expressions (downturned mouth corners, lips apart showing teeth, and lip corner puller) and a sarcastic smirk before speaking suggest a sense of superiority or disdain. The words, 'Isn't he who he is?', convey mockery or ridicule."

Table 15: Comparison of emotional datasets. The table presents a comparative analysis of several key emotional datasets, including DFEW, MER2023, EMER, and MERR. It highlights the unique features and contributions of each dataset, such as the range of emotion categories, availability of multimodal annotations, and dataset size. This comparison underscores the significance of the MERR dataset in advancing multimodal emotion recognition and reasoning research.

| | Sufficient Quantity | Audio Description | Visual Objective Description | Visual Expression Description | Classification Label | **Multimodal Description** |
|---|---|---|---|---|---|---|
| EmoSet [93] | ✔ | ✗ | ✔ | ✔ | ✔ | ✗ |
| EmoVIT [92] | ✔ | ✗ | ✔ | ✔ | ✔ | ✔ |
| DFEW [45] | ✔ | ✗ | ✗ | ✗ | ✔ | ✗ |
| MER2023 [59] | ✔ | ✗ | ✗ | ✗ | ✔ | ✗ |
| EMER [62] | ✗ | ✔ | ✔ | ✔ | ✔ | ✔ |
| MERR (ours) | ✔ | ✔ | ✔ | ✔ | ✔ | ✔ |

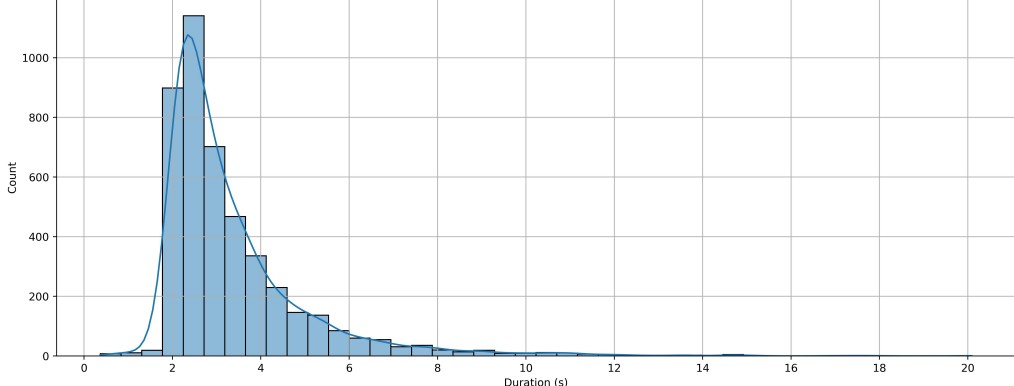

Figure 7: Distribution of video durations in the MERR dataset. The histogram illustrates the range and frequency of video lengths, providing insights into the temporal characteristics of the emotional expressions captured in the dataset. The majority of videos fall within the 2-4 second range, striking a balance between sufficient context and manageable data size for annotation and processing.

# B  Training and Implementation Details

## B.1  Instructions for Multimodal Emotion Recognition

In the multimodal emotion recognition task, the goal is to predict the emotional category label for a given video sample. To guide the model, we use the task identifier [emotion] and provide a set of instructions that prompt the model to classify the emotion displayed in the video. Table 16 presents the list of instructions used for the emotion recognition task. These instructions are designed to be clear, concise, and varied in their phrasing to encourage the model to learn a robust understanding of the task. The instructions ask the model to determine the specific emotion portrayed in the video, choosing from a predefined set of categories: happy, sad, neutral, angry, worried, surprise, fear, contempt, and doubt. By using multiple instructions with slight variations in wording, we aim to prevent the model from overfitting to specific patterns or phrases and instead focus on the underlying task of identifying the emotional state based on the multimodal cues present in the video. During training, these instructions are randomly sampled and paired with the corresponding video samples and their associated annotations. The model learns to attend to the relevant visual, audio, and textual features and map them to the appropriate emotion label based on the provided instruction.

Table 16: List of instructions for multimodal emotion recognition. The table presents a set of carefully crafted instructions that guide the Emotion-LLaMA model in performing emotion recognition tasks. These instructions cover a range of prompts and variations, ensuring the model's ability to understand and respond to different formulations of the emotion recognition problem.

- "Please determine which emotion label in the video represents: happy, sad, neutral, angry, worried, surprise, fear, contempt, doubt."
- "Identify the displayed emotion in the video: is it happy, sad, neutral, angry, worried, or surprise, fear, contempt, doubt?"
- "Determine the emotional state shown in the video, choosing from happy, sad, neutral, angry, worried, surprise, fear, contempt or doubt."
- "Please ascertain the specific emotion portrayed in the video, whether it be happy, sad, neutral, angry, worried, surprise, fear, contempt or doubt."
- "Assess and label the emotion evident in the video: could it be happy, sad, neutral, angry, worried, surprise, fear, contempt, doubt?"

## B.2  Instructions for Multimodal Emotion Reasoning

In addition to recognizing the emotional category, we also want the model to be able to reason about the emotional state based on the available multimodal cues. This task requires a deeper understanding of how the different modalities contribute to the overall emotional interpretation and the ability to explain the reasoning behind the predicted emotion label. To support this task, we use the task identifier [reason] and provide a set of instructions that prompt the model to analyze the multimodal cues and provide a rationale for the predicted emotion. Table 17 shows the list of instructions used for the emotion reasoning task. These instructions ask the model to integrate information from various modalities, such as facial expressions, vocal tone, and the intended meaning behind the spoken words, to infer the emotional state of the person in the video. The model is expected to not only predict the emotion label but also provide a coherent explanation of how the different cues contribute to that prediction. By training the model with these reasoning instructions, we aim to develop its ability to understand the complex interplay between the different modalities and to generate human-interpretable explanations for its predictions. This reasoning capability is crucial for building trust and transparency in the emotion recognition system and facilitating more natural and engaging human-computer interactions. During training, the reasoning instructions are randomly sampled and paired with the video samples and their associated multimodal annotations. The model learns to attend to the relevant cues across modalities, combine them in a meaningful way, and generate a reasoning trace that justifies the predicted emotion label.

Table 17: List of instructions for multimodal emotion reasoning. The table showcases a curated set of instructions designed to elicit emotional reasoning capabilities from the Emotion-LLaMA model. These instructions prompt the model to analyze multimodal cues, generate explanations, and justify its emotional predictions, fostering the development of deep and coherent emotion understanding.

- "Please analyze all the clues in the video and reason out the emotional label of the person in the video."
- "What is the emotional state of the person in the video? Please tell me the reason."
- "What are the facial expressions and vocal tone used in the video? What is the intended meaning behind his words? Which emotion does this reflect?"
- "Please integrate information from various modalities to infer the emotional category of the person in the video."
- "Could you describe the emotion-related features of the individual in the video? What emotional category do they fall into?"

## B.3 Implementation of Emotion-LLaMA Model

The implementation of the Emotion-LLaMA model involves several key components and design choices that enable it to effectively learn and reason about emotions from multimodal data. In this section, we provide a detailed overview of the main implementation details, including the training approach, model architecture, and inference process.

### B.3.1 Multi-Task Learning Approach

One of the core aspects of the Emotion-LLaMA model is its ability to simultaneously learn and perform multiple tasks related to emotion understanding. Specifically, the model is trained using a multi-task learning approach, where the emotion recognition and emotion reasoning tasks are learned in parallel. In the emotion recognition task, the model learns to predict the appropriate emotion label for a given multimodal input, such as a video clip accompanied by audio and text. The model is trained to map the input features to one of the predefined emotion categories, such as happy, sad, angry, or neutral. On the other hand, the emotion reasoning task focuses on generating human-interpretable explanations for the predicted emotion labels. Given a multimodal input, the model learns to analyze the various cues and generate a natural language explanation that justifies the predicted emotion based on the available evidence. By training the model to perform both tasks simultaneously, we allow it to share representations and learn complementary skills across tasks. This multi-task learning approach has several benefits. First, it enables the model to develop a more comprehensive understanding of emotions by learning to recognize the emotional state and explain the reasoning behind it. Second, it encourages the model to learn more robust and generalizable representations by leveraging the commonalities and differences between the two tasks. During training, for each video sample input into the model, we randomly select either the emotion recognition or emotion reasoning task to enhance the model's generalization and robustness.

### B.3.2 Coarse-to-Fine Training Strategy

To facilitate the learning of nuanced and detailed emotional representations, we employ a coarse-to-fine training strategy. This strategy involves two main stages: pre-training on coarse-grained annotations and fine-tuning on fine-grained annotations. In the pre-training stage, the model is trained on the coarse-grained annotations from the MERR dataset. These annotations provide a simple description of emotions, such as visual expression, audio tone. By training on these coarse-grained annotations, the model learns to capture the general emotional tone and develop an initial understanding of the emotional content in the multimodal data. After the pre-training stage, we proceed to the fine-tuning stage, where the model is exposed to the fine-grained annotations from the MERR dataset. These annotations offer more detailed and specific emotion descriptions, such as environmental information, body movements, and the emotional nuances inferred from tone and textual subtitles. By fine-tuning the model on these fine-grained descriptions, we enhance

its emotional understanding and enable it to capture more nuanced expressions and variations in scenarios that closely resemble real-world contexts. The coarse-to-fine training strategy offers several advantages. First, it allows the model to gradually learn more complex and detailed emotional representations, starting from a solid foundation of general emotional understanding. Second, it helps to prevent overfitting and improves the model's generalization ability by providing a hierarchy of emotional annotations to learn from. During the fine-tuning stage, we typically use a smaller learning rate and a shorter training duration compared to the pre-training stage. This allows the model to adapt its representations to the fine-grained labels without drastically changing the learned features from the pre-training stage.

### B.3.3   Modality-Specific Encoders

To effectively process and integrate information from multiple modalities, the Emotion-LLaMA model employs modality-specific encoders. These encoders are designed to extract meaningful features and representations from each modality, capturing the unique characteristics and cues relevant to emotion understanding. For the visual modality, we use a combination of three encoders: MAE (Masked Autoencoders) [36], VideoMAE [84], and EVA (Efficient Video Analysis) [32]. The MAE encoder focuses on capturing local facial features and expressions, which are crucial for recognizing emotions. It learns to reconstruct masked patches of the input frames, enabling it to capture fine-grained details of facial movements and micro-expressions. The VideoMAE encoder, on the other hand, is designed to capture the temporal dynamics and motion patterns in the video data. It learns to predict the masked frames in a video sequence, allowing it to model the evolution of emotional expressions over time. By capturing the temporal context, VideoMAE helps the model to understand the progression and transitions between different emotional states. Finally, the EVA encoder is used to capture the global scene understanding and contextual information in the visual data. It processes the entire video frames and learns to extract high-level features that represent the overall scene and environment. This global context is important for interpreting the emotional state of individuals in relation to their surroundings and the situational factors. For the audio modality, we employ the HuBERT (Hidden-Unit BERT) [39] encoder, which is a state-of-the-art model for speech representation learning. HuBERT is trained on a large corpus of unlabeled speech data and learns to predict the hidden units of a pre-trained teacher model. By doing so, it captures rich vocal representations that encode prosodic features, such as intonation, stress, and rhythm, which are indicative of emotional states. Finally, for the textual modality, we use the LLaMA tokenizer to handle the transcribed speech or dialogue. The LLaMA tokenizer [85] is based on a byte-level Byte Pair Encoding (BPE) [78] algorithm, which allows for efficient and flexible tokenization of the input text. The tokenized text is then passed through the LLaMA model, which learns to capture the semantic and emotional content in the language data.

### B.3.4   Unified Representation Space

To enable the Emotion-LLaMA model to reason about emotions across modalities, we project the outputs from the modality-specific encoders into a unified representation space. This is achieved by applying a linear transformation to the features extracted by each encoder, mapping them to a common embedding dimension. In addition to the projected features, we also introduce learned special tokens that are concatenated with the multimodal embeddings. These special tokens serve as task-specific indicators and provide additional context for the model to distinguish between the emotion recognition and emotion reasoning tasks. The resulting unified representation is then fed into the LLaMA model, which uses its self-attention mechanism to attend to and reason about the multimodal cues in a holistic manner. The LLaMA model learns to capture the interactions and dependencies between the different modalities, enabling it to make informed predictions and generate coherent explanations. By operating in a unified representation space, the Emotion-LLaMA model can effectively integrate and align information from multiple modalities, allowing for cross-modal reasoning and understanding. This unified space facilitates the model's ability to capture the complex dynamics of emotional expressions and generate meaningful insights based on the combined evidence from visual, audio, and textual cues.

### B.3.5   Training Objective and Optimization

The training objective of the Emotion-LLaMA model for both the emotion recognition and emotion reasoning tasks is based on the language modeling loss. This loss function, commonly used in natural language generation tasks, measures the likelihood of generating the ground-truth tokens given the

input multimodal data. By minimizing this loss, the model learns to produce coherent and contextually relevant tokens that align with the provided reasoning examples or emotion categories, effectively capturing the underlying patterns and relationships in the data. To optimize the model parameters during training, we employ the Adam optimizer, a widely adopted and efficient optimization algorithm in deep learning. Adam adapts the learning rate for each parameter based on historical gradients, providing a robust and adaptive optimization process. By leveraging the adaptive learning rates, Adam helps the model converge faster and find better local minima, leading to improved performance and generalization. To train the large language model component of Emotion-LLaMA efficiently, we utilize the Low-Rank Adaptation (LoRA) technique. LoRA significantly reduces the number of trainable parameters while preserving the model's pre-existing world knowledge. By adapting only a small set of low-rank matrices, LoRA allows the model to acquire domain-specific knowledge related to emotions, such as nuances in tone of speech and facial expressions, without overwriting or losing the valuable information learned during pre-training. This technique strikes a balance between efficiency and effectiveness, enabling the model to specialize in the emotion domain while retaining its general language understanding capabilities. For a more detailed of the training objective, optimization process, and the use of LoRA, please refer to the source code[7].

---

[7] https://github.com/ZebangCheng/Emotion-LLaMA

# C  Experiments & Demonstration

## C.1  Evaluation Metrics

To comprehensively assess the performance of the Emotion-LLaMA model on various datasets, we employ a range of evaluation metrics tailored to the specific characteristics and requirements of each dataset. These metrics are designed to provide a fair and thorough evaluation of the model's emotion recognition and reasoning capabilities, taking into account factors such as class imbalance and the importance of both majority and minority classes.

### C.1.1  DFEW Dataset

For the DFEW dataset, we utilize two main evaluation metrics: Weighted Average Recall (WAR) and Unweighted Average Recall (UAR). These metrics are particularly well-suited for imbalanced datasets, where the distribution of samples across different emotion categories is not uniform. WAR focuses on the model's ability to recognize the majority classes, which have a larger number of samples in the dataset. It is calculated as the weighted sum of the recall values for each emotion category, where the weights are determined by the proportion of samples in each category:

$$\text{WAR} = \sum_{i=1}^{N} \left( \frac{n_i}{N} \cdot \text{Recall}_i \right) \tag{4}$$

where $N$ is the total number of emotion categories, $n_i$ is the number of samples in category $i$, and $\text{Recall}_i$ is the recall value for category $i$. On the other hand, UAR ensures that the model's capability to identify the minority classes, which have fewer samples, is not overlooked. It is calculated as the unweighted average of the recall values across all emotion categories:

$$\text{UAR} = \frac{1}{N} \sum_{i=1}^{N} \text{Recall}_i \tag{5}$$

By considering both WAR and UAR, we can assess the model's performance on the majority and minority classes independently, providing a more comprehensive evaluation. A high WAR indicates that the model is effective at recognizing the most prevalent emotions, while a high UAR suggests that the model is capable of identifying even the less frequent emotions accurately. The combination of these metrics offers a balanced assessment, ensuring that it can handle the class imbalance present in the DFEW dataset and provide reliable emotion recognition across all categories.

### C.1.2  MER2023 Dataset

For the MER2023 dataset, we employ the Weighted Average F-score (WAF) as the primary evaluation metric. WAF is a composite metric that combines precision and recall, providing a single value that reflects the model's overall performance. The F-score for each emotion category is calculated as the harmonic mean of precision and recall:

$$F_i = \frac{2 \cdot (\text{Precision}_i \cdot \text{Recall}_i)}{\text{Precision}_i + \text{Recall}_i} \tag{6}$$

where $\text{Precision}_i$ and $\text{Recall}_i$ are the precision and recall values for emotion category $i$, respectively. The WAF is then computed as the weighted average of the F-scores across all emotion categories, with the weights determined by the proportion of samples in each category:

$$\text{WAF} = \sum_{i=1}^{N} \left( \frac{n_i}{N} \cdot F_i \right) \tag{7}$$

where $N$ is the total number of emotion categories, $n_i$ is the number of samples in category $i$, and $F_i$ is the F-score for category $i$. By incorporating both precision and recall, WAF provides a balanced measure of the model's ability to correctly identify emotions while minimizing false positives and false negatives. It takes into account the class imbalance present in the dataset, giving more weight to the performance on the majority classes while still considering the minority classes. WAF offers several advantages over using recall or accuracy alone. It reduces the impact of class imbalance on the evaluation results, ensuring that the model's performance on the minority classes is not overshadowed

by its performance on the majority classes. Additionally, WAF is more robust to noise and outliers, as it considers both the true positive and false positive rates. By using WAF as the evaluation metric for the MER2023 dataset, we can obtain a comprehensive assessment of the Emotion-LLaMA model, taking into account the precision and recall of emotion recognition across all categories.

### C.1.3 EMER Dataset

For the EMER dataset, we employ a unique evaluation approach that leverages the reasoning capabilities of ChatGPT (i.e., gpt-3.5-turbo-16k-0613), a large language model. The evaluation focuses on assessing the quality and coherence of the emotional reasoning provided by the Emotion-LLaMA model. We follow the evaluation prompt template shown in Table 18, where {gt_reason} represents the ground truth emotional description provided in the EMER dataset, and {pred_reason} represents the emotional reasoning generated by the Emotion-LLaMA model or other baseline models. The evaluation prompt is designed to elicit an objective score from ChatGPT based on a set of predefined scoring rules. These rules assess the quality of the generated reasoning in terms of its alignment with the ground truth description, the coherence and logicality of the reasoning process, and the extent to which it captures the relevant emotional cues and evidence.

To ensure a fair and unbiased evaluation, ChatGPT is provided with a clear set of scoring guidelines. These guidelines instruct ChatGPT to consider factors such as the overlap between the predicted and ground truth reasoning, the presence of contradictory or irrelevant information, and the overall clarity and coherence of the generated explanation. ChatGPT then assigns a score to each generated reasoning based on these guidelines, providing a quantitative measure of the model's performance. Along with the numeric score, ChatGPT also provides a brief justification for the assigned score, highlighting the strengths and weaknesses of the generated reasoning.

By leveraging ChatGPT's language understanding and reasoning capabilities, we can obtain a more comprehensive and nuanced evaluation of the Emotion-LLaMA model's performance on the EMER dataset. This approach goes beyond simple metrics like accuracy or F-score and assesses the model's ability to generate coherent and meaningful explanations for the predicted emotions.

Table 18: Prompt for calculating the degree of overlap between emotion-related clues. The table provides a structured prompt template that guides the evaluation of the Emotion-LLaMA model's reasoning capabilities. By comparing the model's generated explanations with the ground truth, the prompt enables the quantitative assessment of the model's ability to identify and articulate relevant emotion-related cues.

---

Below, the "Actual Description" and "Predicted Description" of a character are given. Please follow the steps below to calculate the score for the "Predicted Description". The score should range from 1 to 10. In the end, only output the numerical value of the predicted score along with the reasoning.

1. Summarize the emotional state description of the character from the "Actual Description".

2. Summarize the emotional state description of the character from the "Predicted Description".

3. Calculate the overlap between the "Predicted Description" and the "Actual Description". The higher the overlap, the higher the score.

4. Output format: 'Predicted Score': Predicted Score; 'Reason': Reason

Input:

"Actual Description": {gt_reason}

"Predicted Description": {pred_reason}

Output:

---

## C.2 Additional Results and Analysis

In this section, we present additional results and analysis to further explore the performance and characteristics of the Emotion-LLaMA model. We examine the confusion matrices for the MER2023 and DFEW datasets, discuss the challenges posed by class imbalance, and provide detailed scoring cases from the EMER dataset to showcase the model's emotion reasoning capabilities.

### C.2.1 Confusion Matrices

Figure 8 presents the confusion matrices for the MER2023 and DFEW datasets, providing insights into the model's performance across different emotion categories. In both datasets, the categories "angry," "happy," and "sad" are the most common and distinctive emotions encountered in real-world scenarios. These emotions often have clear and pronounced facial expressions, vocal cues, and language patterns, making them relatively easier for the Emotion-LLaMA model to learn and recognize accurately. The confusion matrices show that the model achieves high accuracy for these prevalent emotion categories. The majority of the samples belonging to "angry," "happy," and "sad" are correctly classified, with minimal confusion between them. This indicates that the model has successfully learned the discriminative features and patterns associated with these emotions and can effectively distinguish between them.

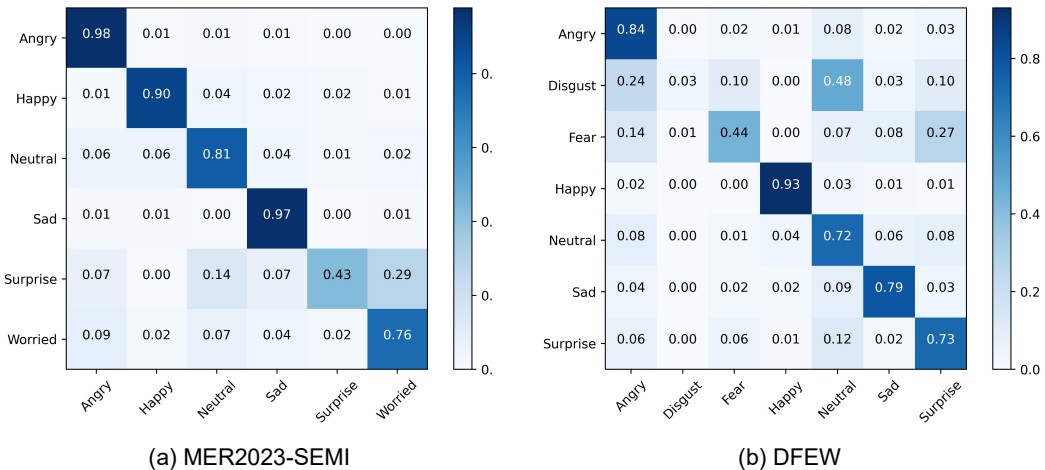

(a) MER2023-SEMI           (b) DFEW

Figure 8: Confusion matrices for multimodal emotion recognition datasets. The figure presents confusion matrices that visualize the performance of the Emotion-LLaMA model on benchmark datasets such as MER2023 and DFEW. The matrices provide insights into the model's classification accuracy, highlighting the challenges and successes in distinguishing between different emotional categories. [Zoom in to view]

However, the confusion matrices also highlight the challenges posed by less frequent emotion categories, such as "disgust" and "fear." Due to the inherent class imbalance present in multimodal emotion recognition datasets, these categories have significantly fewer training samples compared to the more common emotions. The limited availability of training data for "disgust" and "fear" makes it more difficult for the model to learn and generalize the patterns and cues specific to these emotions. As a result, the model may struggle to accurately recognize and classify samples belonging to these categories, leading to higher confusion rates and lower performance compared to the more prevalent emotions. The confusion matrices reveal that a notable proportion of "disgust" and "fear" samples are misclassified as other emotions, such as "angry" or "sad." This suggests that the model may be relying on overlapping or ambiguous features that are shared between these emotions, leading to confusion and misclassification.

Note that addressing the class imbalance issue is a crucial challenge in multimodal emotion recognition. Further research and exploration are needed to develop techniques that can effectively handle the imbalanced distribution of emotion categories and improve the model's performance on less frequent emotions. Potential approaches to mitigate class imbalance include data augmentation techniques, such as oversampling the minority classes or generating synthetic samples, to increase the represen-

tation of underrepresented emotions in the training data. Additionally, employing class-weighted loss functions or adaptive learning strategies that focus on the minority classes during training can help the model learn more robust and generalizable features for these emotions. By tackling the class imbalance problem and improving the model's ability to recognize and classify less frequent emotions accurately, we can enhance the overall performance and practicality of the Emotion-LLaMA model in real-world applications.

### C.2.2 Ablation Study of Hyperparameters

Figure 9 presents additional ablation study results, providing further insights into the impact of different hyperparameters and data settings on the performance of the Emotion-LLaMA model.

One of the key factors investigated in the ablation study is the learning rate. The learning rate determines the step size at which the model's parameters are updated during the training process. It plays a crucial role in the model's convergence and generalization ability. The results in Figure 9 compare the performance of the model across different learning rates. It is observed that the choice of learning rate has a significant impact on the model's performance. Too high a learning rate can lead to unstable training and suboptimal convergence, while too low a learning rate can result in slow convergence and potential overfitting.

The ablation study helps identify the optimal learning rate range for the Emotion-LLaMA model, striking a balance between convergence speed and generalization performance. By carefully tuning the learning rate, we can ensure that the model effectively learns the underlying patterns and features from the training data while avoiding overfitting or underfitting. Another aspect explored in the ablation study is the effect of data proportions on the model's performance. The results compare the model's performance when trained on different subsets of the training data, ranging from smaller fractions to the full dataset. The findings highlight the importance of sufficient training data for the Emotion-LLaMA model to achieve optimal performance. As the proportion of training data increases, the model's performance generally improves, indicating its ability to learn more comprehensive and robust representations of emotions.

Moreover, the ablation study also reveals that the performance gains gradually diminish as the data proportion approaches the full dataset. This suggests that there may be a saturation point beyond which adding more training data yields diminishing returns in terms of performance improvement. Understanding the relationship between data quantity and model performance is crucial for efficient resource allocation and practical deployment considerations. The ablation study results provide guidance on the minimum data requirements for the Emotion-LLaMA model to achieve satisfactory performance and help identify the optimal trade-off between data size and computational resources.

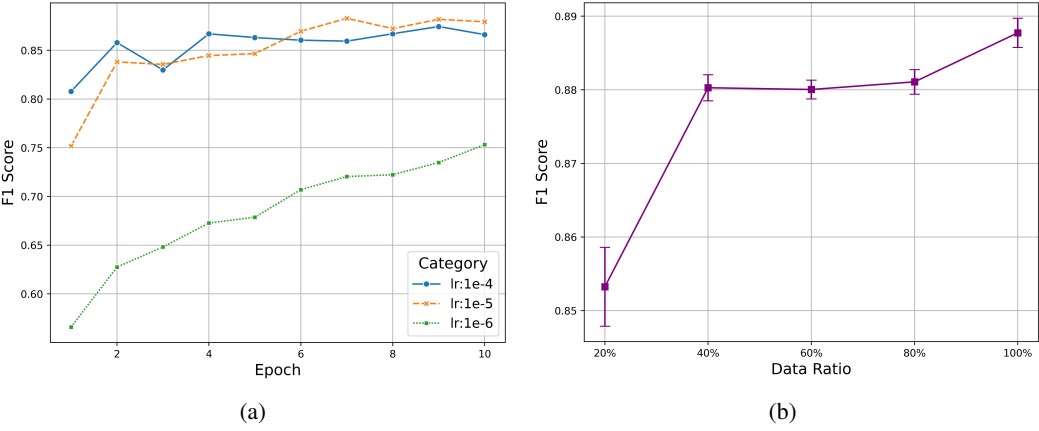

(a)            (b)

Figure 9: Ablation study results. (a) illustrates the impact of different learning rates on the model's performance, while (b) presents the effects of varying data ratios. These experiments provide valuable insights into the optimal hyperparameter settings and data requirements for training the Emotion-LLaMA model effectively. [Zoom in to view]

### C.2.3 MER2024 Challenge

The MER2024 Challenge [60] has recently garnered significant attention for its focus on integrating multimodalities to identify human emotional states. This initiative aims to address the limitations of existing technologies, which often struggle to meet the demands of practical applications. The challenge seeks to advance research in areas such as multimodal modeling of human affect, modality robustness in affect recognition, low-resource affect recognition, human affect synthesis in multimedia, privacy in affective computing, and applications in health, education, and entertainment. The challenge's theme has attracted a large number of researchers from around the globe to discuss recent advancements and future directions for robust multimodal emotion recognition, generating many new insights [9, 29, 74, 79].

The MER2024 Challenge has introduced three distinct tracks: (1) MER-SEMI, which provides over 110,000 unlabeled samples for semi-supervised or self-supervised learning; (2) MER-NOISE, which adds noise to test videos to simulate more realistic variations in modality conditions such as background noise and blurry videos, thereby evaluating system robustness; (3) MER-OV, which requires models to extract richer and more subtle emotions in an open-vocabulary manner, aiming to mitigate the intrinsic subjectivity in the emotion annotation process and the inherent ambiguity of MER. Therefore, the MER-NOISE and MER-OV tracks represent the most realistic expressions of emotions and data distributions, presenting both challenges and practical applications.

The MER-NOISE track emphasizes enhancing noise robustness in emotion recognition systems, as noise is ubiquitous in practical settings. Ensuring clear audio streams and high-resolution video frames can be challenging. This track specifically targets two prevalent types of noise: audio additive noise and image blur, encouraging participants to utilize data augmentation techniques and other innovative methods to bolster the resilience of emotion recognition systems [30]. Due to the integration of multimodal information for reasoning, our proposed Emotion-LLaMA significantly reduces the impact of noise, resulting in outstanding performance in the MER-NOISE track with a weighted average F-score (WAF) of 84.52%, surpassing all other teams. Furthermore, we utilized the predictions from Emotion-LLaMA as pseudo-labels to enhance the performance of Conv-Attention [15]. Ultimately, our team leveraged the robust capabilities of Emotion-LLaMA to secure first place in the MER-NOISE track, exceeding the second and third place teams by 1.47% and 1.65%, respectively. Detailed rankings and scores are shown in Figure 10.

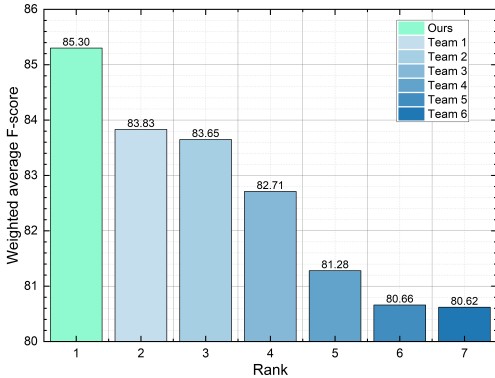

Figure 10: Ranking and Scores of Teams in the MER-Noise Track of MER2024.

The MER-OV track introduces the concept of open-vocabulary emotion recognition, addressing the subjectivity and ambiguity inherent in emotion labeling. Traditional datasets often limit label spaces to a few discrete categories, relying on multiple annotators and majority voting, which can overlook valid but non-candidate or minority labels. Participants generate a diverse array of labels, aiming for a more nuanced and accurate representation of emotional states. [33, 98]. As shown in Table 4, Emotion-LLaMA demonstrates greater generalization capabilities compared to other multimodal large language models.

### C.2.4 Detailed Scoring Cases

To showcase the emotion reasoning capabilities of the Emotion-LLaMA model, we present detailed scoring cases from the EMER dataset in Table 19 and Table 20.

In these tables, we compare the emotion reasoning results generated by the Emotion-LLaMA model with those of other multimodal large language models. The evaluation is performed using the prompts from Table 18, which assess the quality and coherence of the generated reasoning against the ground truth emotional descriptions. The scoring cases demonstrate the superiority of the Emotion-LLaMA model in accurately identifying and interpreting the emotional cues present in the multimodal data. The model successfully captures the facial details, vocal tone, and linguistic content, and integrates them to arrive at the correct emotion labels.

In Table 19, the Emotion-LLaMA model accurately identifies the subtle facial expressions and the slightly raised voice of the speaker, indicating a sense of doubt or uncertainty. By combining these cues with the content of the spoken dialogue, the model generates a coherent and logical reasoning that aligns well with the ground truth description. Similarly, in Table 20, the Emotion-LLaMA model correctly recognizes the facial cues of raised eyebrows and a slight frown, suggesting a state of surprise or disbelief. The model also takes into account the emphatic tone of the speaker and the context of the conversation to infer the underlying emotion accurately.

In contrast, the other multimodal large language models struggle to provide accurate and coherent reasoning in these cases. They may focus on irrelevant details, generate inconsistent explanations, or fail to capture the nuances of the emotional expressions. The detailed scoring cases highlight the effectiveness of the Emotion-LLaMA model in performing multimodal emotion reasoning.

Table 19: Comparison of reasoning results, scores, and evaluation criteria between Emotion-LLaMA and other popular MLLMs for "sample_00002348". The table presents a detailed analysis of the emotional reasoning capabilities of different models on a specific data sample. It highlights the superiority of Emotion-LLaMA in generating accurate, coherent, and well-justified explanations, as evidenced by the higher scores and more comprehensive evaluation criteria.

| An Example of Multimodal Emotion Reasoning | |
| --- | --- |
| |  |
| Subtitle (En) | Ah. Wei Wei is pregnant. |
| Ground Truth | In the video, she found out that her friend is pregnant and he was very surprised. She spoke faster, raised her voice, and suddenly sat up from the sofa. She opened her mouth wide, widened her eyes, and raised her eyebrows. Therefore, it can be inferred that her emotional state is surprise. |
| PandaGPT | This person's emotional state appears to be positive and happy, as they are smiling while sitting on the bed. The presence of a pregnancy subtitle indicates that they are expecting a child, which is likely the source of their happiness and excitement. |
| Criteria | The prediction mentions the character sitting on the bed smiling, and a subtitle about pregnancy indicates that they are expecting a child, suggesting a positive and happy emotional state. Although the prediction does not explicitly mention the character's surprised emotional state, there is some overlap between the described positive and happy emotions and the surprised emotion in the actual description, leading to a higher prediction score. |
| Score | 7.0 |
| VideoChat-Embed | Based on the video, it can be inferred that this person is clearly unhappy, as they are on the phone, shaking their head, and displaying a pained expression. The fact that they are crying indicates they are deeply troubled or under significant stress. The video mentions a pregnant woman, so this person may be upset about a recently born baby. However, without more information about the phone conversation or context, it is difficult to determine the exact emotional state. |
| Criteria | The "Predicted Description" about the character's emotional state has some overlap with the "Actual Description", as both mention their emotional condition. However, the "Predicted Description" also mentions other possible emotional states, such as unhappiness and pain, which do not fully align with the surprise described in the "Actual Description". Therefore, the score is 6. |
| Score | 6.0 |
| Valley | From the video, we can infer that this woman is in a calm and relaxed emotional state. She sits on the bed, occasionally smiling while talking on the phone. Her facial expressions and body language indicate that she feels comfortable and is enjoying the conversation. Additionally, the woman is wearing a white shirt, which contributes to a casual and relaxed appearance. |
| Criteria | The "Actual Description" describes the male protagonist's emotional state as surprised, whereas the "Predicted Description" describes the female protagonist's emotional state as calm and relaxed. The emotional states described are completely different, with very low overlap, therefore the prediction score is low. |
| Score | 3.0 |
| Video-ChatGPT | From the video, we can see that the individual is happy and excited about the news of the slight pregnancy. The woman is on the phone with a smile on her face, which indicates her happiness and excitement. |
| Criteria | The "Actual Description" describes the character's emotional state as surprised, while the "Predicted Description" describes it as happy and excited. Although the emotional states are not exactly the same, both are positive emotional states and are based on observations from the video. Therefore, it can be considered that there is some degree of overlap between the two, and the prediction score is 6. |
| Score | 6.0 |
| Video-LLaMA | The video shows a young woman sitting on a sofa, talking on her mobile phone. From the description, we can see that she is a young Asian female wearing a white dress. The lady is looking at the camera, which suggests that she is on a call. She appears to be laughing or smiling, indicating that she is in a good mood. The background features a white ceiling with a black clock, suggesting that the lady is in a comfortable setting. Overall, the video portrays a happy woman enjoying a phone conversation while relaxing on the sofa. |
| Criteria | According to the "Actual Description," the character displays a surprised emotional state, whereas the "Predicted Description" describes the character's emotional state as happy and relaxed. Although there is no obvious overlap between the two, based on the descriptions, it can be speculated that the character might experience some pleasant surprises during the phone conversation. Therefore, a prediction score of 6 is given. |
| Score | 6.0 |
| Emotion-LLaMA | In the video, a woman is seen sitting on a couch, displaying facial expressions and body language that suggest surprise and alertness. Her raised eyebrows, strong emotional tone, and quickened breathing further reinforce this interpretation. Therefore, it is inferred that her emotional state is surprise. |
| Criteria | The predicted description closely matches the actual description regarding the emotional state of the person, including consistent descriptions of facial expressions, tone of voice, and breathing. Therefore, the score is high. |
| Score | 9.0 |

Table 20: Comparison of reasoning results, scores, and evaluation criteria between Emotion-LLaMA and other popular MLLMs for "sample_00006957". Similar to the previous table, this table showcases the comparative performance of Emotion-LLaMA and other MLLMs on another data sample. It demonstrates the model's ability to capture nuanced emotional cues and provide compelling explanations, outperforming other approaches in reasoning quality and alignment with human judgments.

**An Example of Multimodal Emotion Reasoning**

| | |
|---|---|
| Subtitle (En) | I have to criticize you. Anyway, Mr. Wang is your teacher. How can you talk to Mr. Wang like that? |
| Ground Truth | In the video, he is dissatisfied and angry with the other person's disrespectful behavior towards Teacher Wang, and is currently criticizing the other person. The volume has been raised and the speech speed has also increased in the backend of the video. Therefore, it is inferred that his emotional state is anger. |
| PandaGPT | The individual appears to be depressed or frustrated as they sit in a chair and talk on the phone with someone, hand on their forehead. The subtitles suggest they are criticizing their teacher, Mr. Wang, and expressing dissatisfaction with his teaching methods. This implies that the person may be facing some challenges or disagreements with their teacher, leading to their negative emotional state. |
| Criteria | The predicted description of the emotional state overlaps somewhat with the actual description of the emotional state, both mentioning dissatisfaction and criticism towards the teacher. Although the predicted description refers to feelings of depression or frustration, which differ from the anger mentioned in the actual description, both express dissatisfaction with the teaching methods. Therefore, the prediction of the emotional state in the predicted description can be considered accurate. Due to the high degree of overlap, a high score is awarded. |
| Score | 7.0 |
| VideoChat-Embed | From the video, we can infer the person's emotional state by observing their facial expressions and tone of voice. The individual appears confident and assertive during the conversation with the teacher, indicating that they are neither overly anxious nor uncomfortable. On the other hand, the teacher feels frustrated by the student's lack of cooperation and professionalism during the dialogue. |
| Criteria | The predicted description mentions that the individual appears confident and assertive during the conversation with the teacher, which partially overlaps with the actual description where the character is dissatisfied and angry with the disrespectful behavior towards Mr. Wang. Additionally, the predicted description also notes the teacher's frustration due to the student's lack of cooperation and professionalism in the dialogue, which aligns to some extent with the emotional state described in the actual context. Therefore, given the considerable overlap between the predicted and actual descriptions, a score of 7 is assigned. |
| Score | 7.0 |
| Valley | From the video, we can infer that the man is likely in a professional setting, possibly a doctor's office, as he is dressed in a white shirt and tie. He is also looking at a clipboard, which suggests he might be reviewing some medical records or notes. Moreover, the video shows the man conversing with a woman, who could be a patient or colleague, and there is a computer and a chair in the room. Based on these visual cues, we can deduce that the man's emotional state is likely calm, focused, and professional, as he is engaged in a conversation and appears to be handling medical or work-related tasks. |
| Criteria | The predicted description of the emotional state partially overlaps with the actual description, as both mention the man's emotional state. In the actual description, he is angry, while in the predicted description, he is calm, focused, and professional. Although the two descriptions are not entirely consistent, both involve emotional states, so a certain score can be justified. |
| Score | 7.0 |
| Video-ChatGPT | The person in the video appears to be an older man who is conversing with a younger woman. He is dressed in a white coat and seated across from her. The man talks to the woman in a calm manner, while she listens attentively. Based on these visual cues, we can infer that the man is likely a doctor or a professional, and the woman is either his patient or colleague. The man converses with the woman in a professional manner, indicating that he is providing her with medical advice or discussing professional matters. Solely based on visual clues, it is not possible to determine the man's emotional state, but it is clear that he is engaging in a conversation with the woman. |
| Criteria | Based on the emotional state described in the "Actual Description," where the character feels dissatisfied and angry, and the "Predicted Description," which does not mention the character's emotional state, it is impossible to determine the character's true emotional state. The overlap is minimal, hence a score of 5 is assigned. |
| Score | 5.0 |
| Video-LLaMA | This video showcases a man and a woman in a professional office setting. The woman is speaking to the man, who appears to be a doctor, discussing a case she is handling. They are seated next to a table covered with documents. The woman seems unhappy or stressed while speaking. Throughout the video, the sound of a clock ticking can be heard in the background. The scene is shot in a medium frame with both individuals sitting by the table ... |
| Criteria | According to the "Actual Description," the character is angry, whereas the "Predicted Description" does not clearly define the character's emotional state but merely describes their conversation in a professional office setting. Therefore, there is a low degree of overlap between the predicted and actual descriptions, resulting in a score of 6. |
| Score | 6.0 |
| Emotion-LLaMA | In the video, a doctor sits at a desk, displaying facial expressions—a furrowed brow, raised upper eyelids—that suggest anger or frustration. Along with a tone that implies annoyance or irritation, his words, "How can you talk to Mr. Wang like that?" convey a sense of disapproval and disappointment. This indicates that the person in the video is experiencing anger due to the other person's misbehavior. Therefore, it is inferred that his emotional state is anger. |
| Criteria | The predicted description of the character's emotional state highly overlaps with the actual description, including aspects such as facial expressions, tone, and words, all conveying feelings of anger. Due to this significant overlap, a high prediction score is awarded. |
| Score | 9.0 |

### C.3  Demonstration of Emotion-LLaMA

To showcase the capabilities and potential applications of the Emotion-LLaMA model, we have developed an interactive demo that allows users to experience its emotion recognition and reasoning functionalities firsthand. The demo provides a user-friendly interface for inputting multimodal data, such as images, videos, and text, and receiving real-time emotion predictions and explanations.

#### C.3.1  General Task Performance

Figure 12 illustrates the demo's performance on general tasks, such as face detection and question answering. These tasks demonstrate the versatility and robustness of the Emotion-LLaMA model beyond its core emotion understanding capabilities. In the face detection task, the demo takes an input image and accurately identifies and localizes the faces present in the image. The model's ability to detect faces is crucial for subsequent emotion recognition, as it allows the system to focus on the relevant regions of interest and extract facial features effectively. The demo also showcases the model's question-answering capabilities. Given a question and an associated image or video, the Emotion-LLaMA model can generate accurate and contextually relevant answers. By leveraging its multimodal understanding, the model can reason about the visual content and provide informative responses to user queries. These general task examples highlight the Emotion-LLaMA model's ability to handle a wide range of tasks that involve visual perception, language understanding, and reasoning. The model's performance on these tasks demonstrates its potential to be integrated into various applications, such as intelligent assistants and human-computer interaction interfaces.

#### C.3.2  Multimodal Emotion Recognition and Reasoning

Figure 13 focuses on the core functionalities of the Emotion-LLaMA model: multimodal emotion recognition and reasoning. The demo allows users to input various combinations of visual, audio, and textual data and receive real-time emotion predictions and explanations. In the emotion recognition task, the user can provide an image or video depicting a person's facial expressions, body language, and contextual cues. The Emotion-LLaMA model processes the visual input, extracting relevant features and patterns, and predicts the most likely emotion category based on its trained knowledge. The demo displays the predicted emotion label along with the corresponding confidence score, indicating the model's level of certainty in its prediction. This information helps users interpret the model's output and assess the reliability of the emotion recognition result.

In addition to emotion recognition, the demo enables users to explore the model's emotion reasoning capabilities. By providing a multimodal input, such as a video clip accompanied by audio and text, users can request the Emotion-LLaMA model to generate a natural language explanation for its predicted emotion. The model analyzes the multimodal data, considering the facial expressions, vocal cues, and linguistic content, and generates a coherent and human-like explanation for the identified emotion. The generated explanation highlights the specific cues and patterns that contributed to the model's prediction, providing insights into its reasoning process. The emotion reasoning feature of the demo is particularly valuable for applications that require transparent and interpretable emotion understanding. By providing explanations alongside the predicted emotions, the Emotion-LLaMA model enables users to gain a deeper understanding of the factors influencing the model's decisions and fosters trust in the system's outputs.

The demo also allows users to compare the Emotion-LLaMA model's performance with other baseline models or human annotations. By presenting the emotion predictions and explanations from multiple sources side by side, users can assess the model's accuracy, coherence, and alignment with human judgments. This comparative analysis feature of the demo facilitates the evaluation and validation of the Emotion-LLaMA model's performance in real-world scenarios. It provides a platform for researchers, developers, and end-users to explore the model's strengths, identify areas for improvement, and gather insights for further refinement and adaptation.

#### C.3.3  Potential Applications and Impact

The Emotion-LLaMA demo serves as a powerful showcase of the model's capabilities and highlights its potential applications across various domains. Some of the key areas where the Emotion-LLaMA model can make a significant impact include:

1. **Affective Computing:** The Emotion-LLaMA model can be integrated into affective computing systems to enable more natural and empathetic human-computer interactions. By recognizing and responding to users' emotions, these systems can provide personalized and emotionally intelligent experiences.

2. **Mental Health Assessment:** The model's ability to recognize and reason about emotions from multimodal data can be leveraged in mental health assessment tools. It can assist mental health professionals in analyzing patient data, identifying emotional patterns, and providing objective insights to support diagnosis and treatment planning.

3. **Sentiment Analysis:** The Emotion-LLaMA model can be applied to sentiment analysis tasks, such as social media monitoring, customer feedback analysis, and brand reputation management. By accurately identifying the emotions expressed in user-generated content, businesses can gain valuable insights into public opinion and make data-driven decisions.

4. **Emotion-Aware Robotics:** The model can be integrated into robotics systems to enable more natural and engaging human-robot interactions. By recognizing and responding to human emotions, emotionally intelligent robots can provide more empathetic and personalized assistance in various settings, such as healthcare, education, and customer service.

5. **Emotional Intelligence Training:** The Emotion-LLaMA model can be used to develop emotional intelligence training programs and tools. By exposing individuals to a wide range of emotional expressions and providing explanations for the underlying cues and patterns, the model can help users enhance their emotional awareness and empathy skills.

These are just a few examples of the potential applications and impact of the Emotion-LLaMA model. As the field of multimodal emotion understanding continues to advance, the model's capabilities can be further extended and adapted to address new challenges and opportunities across various domains.

For a hands-on experience with the Emotion-LLaMA model, you can explore the online demo on Hugging Face[8], which allows you to input multimodal data samples and observe the model's emotion recognition and reasoning capabilities in action. The demo provides an interactive interface to visualize the input data, the predicted emotion labels, and the generated explanations, giving you a comprehensive understanding of the model's performance and potential applications. As the demo continues to evolve and the Emotion-LLaMA model is further refined, we envision its increasing adoption and integration into real-world systems, empowering to harness the power of multimodal emotion understanding for a wide range of purposes.

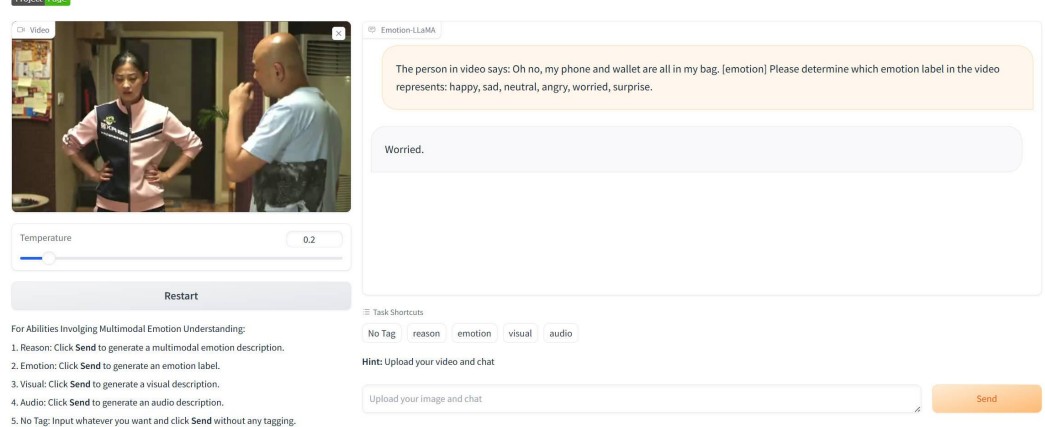

Figure 11: Online demo interface of the Emotion-LLaMA model. The figure presents a screenshot of the interactive web-based demo, showcasing the model's capabilities in multimodal emotion recognition and reasoning. Users can input various combinations of visual, audio, and textual data and receive real-time emotion predictions and explanations, facilitating intuitive exploration and evaluation of the model's performance.

---

[8]https://huggingface.co/spaces/ZebangCheng/Emotion-LLaMA

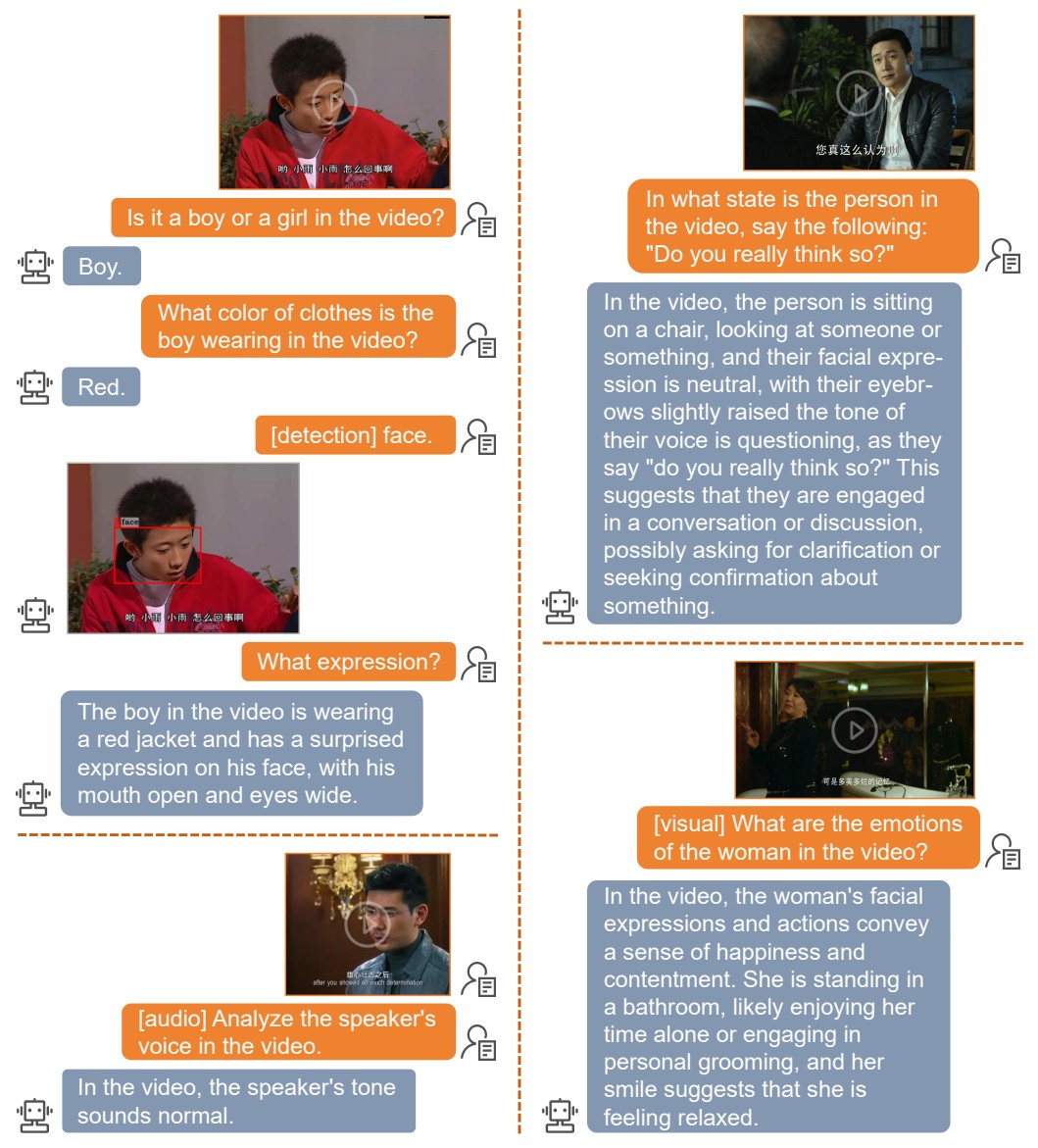

Figure 12: Detailed examples of general tasks performed by the Emotion-LLaMA model. The figure illustrates the model's versatility and robustness in handling tasks beyond emotion recognition, such as face detection and question answering. These examples highlight the model's ability to process and understand visual and textual information, enabling its application in a wide range of scenarios.

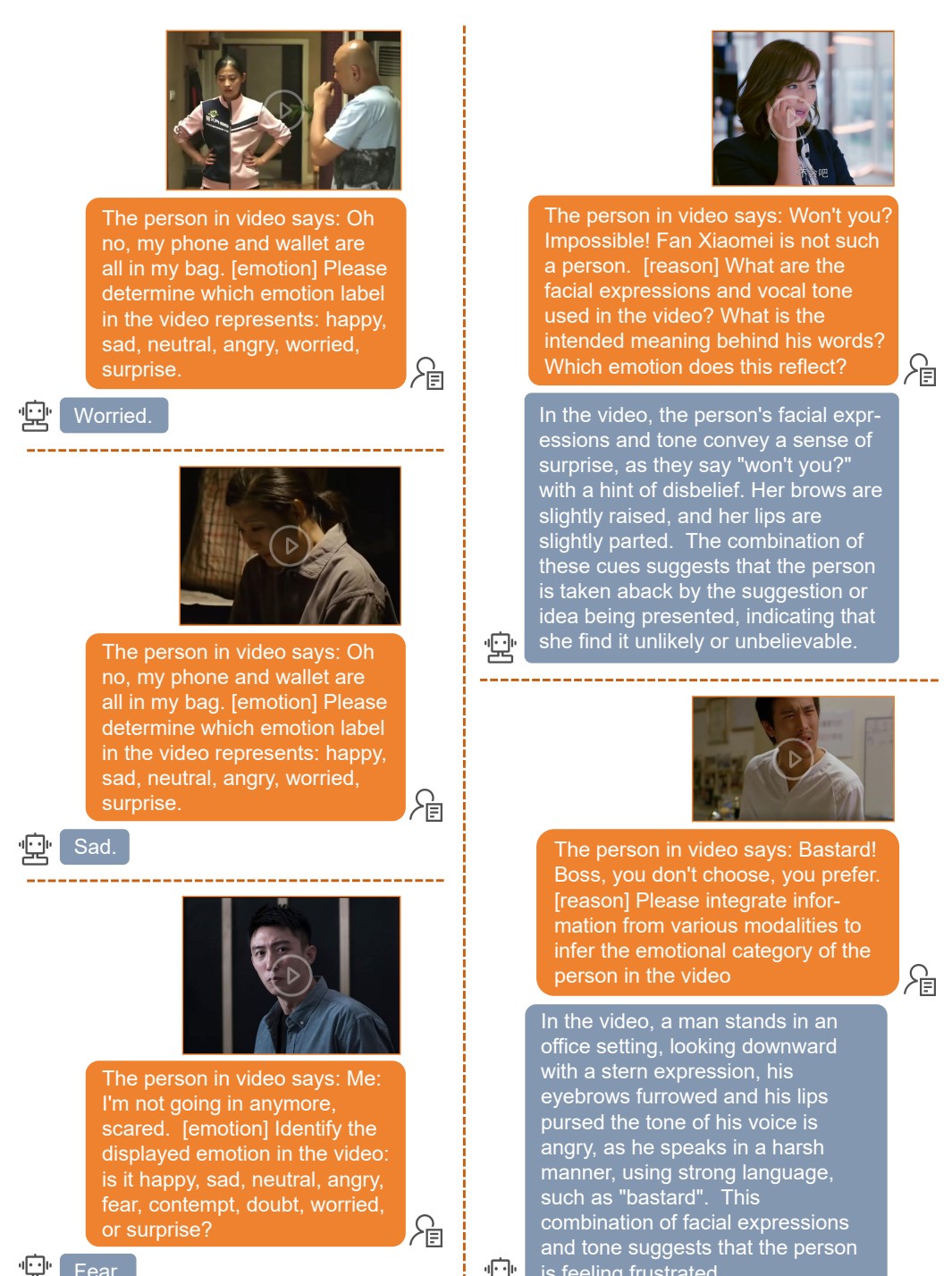

Figure 13: Detailed examples of multimodal emotion recognition and reasoning performed by the Emotion-LLaMA model. The figure showcases the model's core capabilities in accurately identifying emotions from multimodal data and generating human-like explanations for its predictions. These examples demonstrate the model's proficiency in capturing subtle emotional cues, integrating information across modalities, and providing meaningful insights into its decision-making process.

