# OpenReview forum: "Emotion-LLaMA: Multimodal Emotion Recognition and Reasoning with Instruction Tuning"
_NeurIPS.cc/2024/Conference — NeurIPS 2024 poster_

### Official Review · Reviewer_646t · 2024-06-22

**Soundness:** 2
**Presentation:** 2
**Contribution:** 1
**Rating:** 5
**Confidence:** 4

**Summary:**

This paper implements multimodal emotion recognition and reasoning by fine-tuning the LLaMA model with instructions. It is trained on a large-scale dataset, fine-tuned, and tested on three datasets.

**Strengths:**

The global, temporal and local features of the video modality are considered, and LoRA fine-tuning (all tokens), Prompt fine-tuning (text modality) and supervised fine-tuning (linear layer) are used at the same time.

**Weaknesses:**

The innovation is limited and only existing methods are used. The key problems are not solved.

**Questions:**

1. The complex interaction relationship between modalities is not considered.

2. The process is complicated, and the computing resource requirements are high.

3. The effect of Prompt Tuning and Instruction Tuning is highly dependent on the design of prompt words and instructions. If the prompt words or instructions are not accurate or comprehensive enough, it may impact the performance of the model.

4. The lack of reproducibility: the anonymous Github link provided by the author is empty.

**Limitations:**

Yes

---

> ### Author Rebuttal · Authors · 2024-08-06
>
> Thank you for your insightful review. We've addressed your points carefully and will incorporate these clarifications in our revision.
>
> **Q1: The complex interaction relationship between modalities is not considered.**
>
> We appreciate your attention to the interaction between modalities. However, we respectfully suggest that the premise of this question may not fully align with our approach and findings:
>
> 1. Early works in multimodal large language models (MLLMs) often used the Q-Former proposed in BLIP2, involved complex two-stage pre-training, which could lead to information loss.
>
> 2. Direct feature mapping: Emotion-LLaMA directly maps visual and audio features into the textual semantic space. This approach preserves unique characteristics from each modality (e.g., facial micro-expressions, vocal tones) while allowing for their interaction in the unified semantic space.
>
> 3. Experimental validation: Our experiments in the MER2024 competition demonstrate the effectiveness of this approach:
>
>    | Modality Combination | Pre-alignment F1 | Post-alignment F1 |
>    |----------------------|------------------|-------------------|
>    | Audio only (HuBERT)  | 0.7277           | 66.18             |
>    | Video only (CLIP)    | 0.6673           | 69.07             |
>    | Text only (Baichuan) | 0.5429           | 56.85             |
>    | All modalities       | 80.91            | 73.01             |
>
> These results suggest that complex alignment methods may benefit weaker modalities but can also lead to information loss (a 7.9% decrease in fusion scores post-alignment).
>
> **Q2: The process is complicated, and the computing resource requirements are high.**
>
> We appreciate your concern about computational efficiency. However, we believe our method is relatively resource-efficient:
>
> 1. Parameter-efficient tuning: We use LoRA, resulting in only 34 million trainable parameters (0.495% of the total).
> 2. Hardware requirements: All work was completed using only 4 A100 (40G) GPUs, which is modest compared to many LLM pre-training efforts.
> 3. Inference efficiency: Emotion-LLaMA requires only a single A10 or A100 GPU for rapid inference.
>
> **Q3: Dependence on Prompt and Instruction Design**
>
> A3: Designing effective prompts and instructions is indeed crucial for harnessing the powerful reasoning abilities of LLMs. In this work, we have made significant contributions by designing various training tasks and different instructions for Emotion-LLaMA to enhance its robustness and generalization capabilities. We continuously refine our prompts and instructions based on iterative testing and validation to ensure they are accurate and comprehensive. The demo in the anonymous repository showcases Emotion-LLaMA's excellent performance, demonstrating that it can provide correct answers regardless of whether the instructions are previously learned or new, highlighting the robustness and innovation of our design.
>
> **Q4: The lack of reproducibility: the anonymous GitHub link provided by the author is empty.**
>
> A4: We apologize for the inconvenience you experienced. This issue may have been due to a temporary network problem. We have verified that the anonymous repository is accessible and contains all necessary files for reproducibility. Our Emotion-LLaMA has been successfully reproduced by other researchers, who have praised its performance in the demo. We encourage you to access the repository again to review the open-source MERR dataset, the training process, and the code for Emotion-LLaMA. We are committed to ensuring that all materials are available and easily accessible for reproducibility.
>
> **Q5: The innovation is limited, and only existing methods are used. The key problems are not solved.**
>
> A5: We respectfully disagree with this assessment. Our work addresses critical challenges in multimodal emotion recognition and reasoning:
>
> 1. **Data scarcity**: Existing datasets predominantly consist of image-text pairs, lacking dynamic expression descriptions and audio components. Manual annotation of real-world multimodal samples is prohibitively expensive. To address this, we introduced the MERR dataset, comprising 28,618 coarse-grained and 4,487 fine-grained automatically annotated samples across diverse emotion categories. This dataset significantly advances the field by providing rich, multimodal emotion-related instruction-following data.
>
> 2. **Real-world applicability**: Existing MLLMs struggle with accurate emotion understanding in real-world scenarios. Emotion-LLaMA, instructed-tuned on the MERR dataset, has demonstrated excellent performance in both controlled and real-world conditions. Our first-place win in the MER2024 competition's MER-Noise track underscores its robustness in noisy, real-world environments.
>
> 3. **Generalization**: In the MER-OV track of the [MER2024 competition][1], Emotion-LLaMA significantly outperformed other MLLMs, including GPT-4V, by improving average accuracy and recall by 8.52%. This showcases its superior generalization capabilities across various emotion understanding tasks.
>
> 4. **Comprehensive evaluation**: Extensive experimental results demonstrate Emotion-LLaMA's outstanding multimodal emotion understanding capabilities across multiple benchmarks and real-world scenarios.
>
> 5. **Reproducibility and accessibility**: We have open-sourced both the MERR dataset and the code for Emotion-LLaMA, and provided an online demo in our [anonymous repository][2]. This facilitates further research and practical applications in the field of multimodal emotion recognition and reasoning.
>
> While we acknowledge that there is always room for improvement, we believe these contributions offer valuable insights into addressing challenges in multimodal emotion understanding. We welcome further discussion on how we can enhance our approach to better solve key problems in the field.
>
> [1]: https://zeroqiaoba.github.io/MER2024-website/
> [2]: https://anonymous.4open.science/r/Emotion-LLaMA/

---

> ### Author Response · Authors · 2024-08-10
> **Response to Reviewer 646t**
>
> Dear Reviewer,
>
> Thank you for your insights. We have made the following revisions to address your concerns:
>
> 1. We clarified the interaction between modalities in our approach, demonstrating the effectiveness of our method.
> 2. We discussed the computational efficiency of our approach and how we ensured that the process is resource-efficient.
> 3. We ensured the reproducibility of our work by verifying that all necessary files are accessible in our anonymous repository.
> 4. We highlighted the innovative aspects of our methodology, addressing the challenges in multimodal emotion recognition.
>
> We hope these updates address your concerns. We appreciate any further feedback you might have.
>
> Best regards,
> The Authors

---

> ### Author Response · Authors · 2024-08-12
> **Follow-Up on Revisions and Inquiry on Additional Concerns**
>
> Dear Reviewer 646t,
>
>
> Thank you for your thorough review and for raising the score, which we greatly appreciate. As we approach the rebuttal deadline, we wanted to check if there are any remaining concerns or questions that we could address. Your feedback has been invaluable, and we are committed to making any further necessary improvements.
>
> Please let us know if there is anything else we should consider.
>
>
> Best regards,
>
> The Authors

---

> ### Author Response · Authors · 2024-08-13
> **Follow-Up on Revisions and Interaction**
>
> Dear Reviewer 646t,
>
> Thank you for your thorough review and for raising the score, which we greatly appreciate. We have invested a significant amount of effort into this work, and your feedback has been instrumental in guiding our revisions.
>
> As we approach the rebuttal deadline, we wanted to ensure that all your concerns have been adequately addressed. We would also like to invite you to try out our demo, available in the anonymous repository. Our work has already attracted considerable attention, leading to a high number of visits to the demo, which has significantly increased the maintenance costs. Despite this, we have kept it running during the review process to provide valuable insights into the practical application of our methods.
>
> If there are any remaining questions or additional feedback you could provide, we would be more than happy to address them.
>
> Best regards,
>
> The Authors

---

### Official Review · Reviewer_V8m9 · 2024-07-10

**Soundness:** 3
**Presentation:** 3
**Contribution:** 3
**Rating:** 5
**Confidence:** 3

**Summary:**

The paper introduces a multimodal large language model, named Emotion-LLaMA for emotional state understanding. The authors use open-source tools to collect and annotate a dataset, named MERR for model pre-training. Then they perform instruction-tuning on downstream datasets for emotion recognition and emotion reasoning. Extensive experiments are conducted and demonstrate the promising performance of the proposed approach.

**Strengths:**

1. Extensive experiments are conducted and the method achieves SOTA performance on various dataset for emotion recognition and reasoning.
2. Straightforward visualizations are presented.

**Weaknesses:**

1. The paper claims the MERR dataset as a core contribution. However, there is no systematic evaluation for the label quality of the dataset. I understand that pre-training on this dataset improves the downstream performance and thus its validity can be shown to some extent. However, this may be due to the diversity of the unlabeled data instead of the automatically generated label.

2. Table 3, with audio and video inputs, Emotion-LLaMA’s performance is worse/close to the baseline (VAT).

**Questions:**

1. How do you pre-train Emotion-LLaMA? Is it supervised learning with coarse-grained emotion labels?

2. Do you plan to release the MERR dataset?

3. Table 2 fine-tuning, why does Emotion-LLaMA get worse performance on Disgust than MAE-DFER and VideoMAE?

**Limitations:**

Please refer to weaknesses

---

> ### Author Rebuttal · Authors · 2024-08-06
>
> We appreciate your insightful review. Your feedback has been valuable in improving our work. We've addressed each point below and will incorporate these enhancements in our revision.
>
> **Q1: How do you pre-train Emotion-LLaMA? Is it supervised learning with coarse-grained emotion labels?**
>
> A1: Yes, we use automatically generated labels and descriptions as answers for supervised learning. Instruction tuning of Emotion-LLaMA involves two main stages, utilizing both coarse-grained and fine-grained data from our MERR dataset. Here's a detailed breakdown of our training process:
>
> 1. **Data Preparation:**
>    - Coarse-grained data: 28,618 samples
>    - Fine-grained data: 4,487 samples
>
> 2. **Feature Extraction:**
>    - Audio features: Extracted using HuBERT-Chinese
>    - Visual features: Extracted using a combination of MAE, VideoMAE, and EVA2
>    - Text tokens: Processed directly by BERT’s Tokenizer
>
> 3. **Prompt Construction:**
>    We construct prompts relevant to the training task, examples of which can be found in **Tables 11 and 12** of our paper.
>
> 4. **Instruction Template:**
>    We use the following template to form our instructions:
>    ```
>    [INST] < AudioFeature > < VideoFeature > [Task Identifier] Prompt [/INST]
>    ```
>    Where:
>    - `< AudioFeature >` and `< VideoFeature >` are the extracted features
>    - `[Task Identifier]` specifies whether it's a recognition or reasoning task
>    - `Prompt` is the constructed prompt for the specific task
>
> 5. **Answer Preparation:**
>    For each instruction, we prepare corresponding answers:
>    - For recognition tasks: Emotion category labels
>    - For reasoning tasks: Detailed emotion descriptions
>
> 6. **Training Procedure:**
>
>    a. **Stage 1 - Coarse-grained Pretraining:**
>    - Data: 28,618 coarse-grained samples
>    - Epochs: 30
>    - Batch size: 4
>    - Learning rate: 1e-5 with cosine decay
>
>    b. **Stage 2 - Fine-grained Tuning:**
>    - Data: 4,487 fine-grained samples
>    - Epochs: 10
>    - Batch size: 4
>    - Learning rate: 1e-6 with cosine decay
>
> 7. **Implementation:**
>    The specific code (train.py, train_configs/Emotion-LLaMA_finetune.yaml) and more implementation details (README.md: Setup, Training) are available in our anonymous repository (https://anonymous.4open.science/r/Emotion-LLaMA). Detailed descriptions of the training process and feature extraction can be found in **Sec. 4.2** of our paper.
>
> **Q2: Do you plan to release the MERR dataset?**
>
> A2: Yes, the MERR dataset is available on GitHub as part of our submission. You can access and view the MERR dataset (README.md: MERR Dataset) in the anonymous repository mentioned in the paper.
>
> **Q3: The improvement might be due to the diversity of unlabeled data rather than the automatically generated labels.**
>
> A3: While the MER2023 dataset contains many unlabeled video samples, MLLMs require instruction datasets for training and cannot be directly trained with unlabeled samples. We built the MERR dataset with 28,618 coarse-grained and 4,487 fine-grained annotated samples with rich emotional descriptions. Instruction tuning based on these labels and descriptions significantly enhanced Emotion-LLaMA's emotional understanding capability. The improvement is due not only to the diversity of the unlabeled data but also to the quality of the automatically generated labels and descriptions.
>
> **Q4: Why does Emotion-LLaMA perform worse on 'Disgust' than MAE-DFER and VideoMAE?**
>
> A4: As shown in **Tab. 2**, almost all existing MLLMs perform poorly in recognizing 'disgust', often with zero accuracy. This is likely due to the scarcity of 'disgust' samples in current datasets. We also suspect LLMs may have safety restrictions related to 'disgust', contributing to the low accuracy. We plan to collect more samples to enrich the MERR dataset and further explore this issue to improve Emotion-LLaMA's ability to recognize new categories.
>
> **Q5: In Table 3, with audio and video inputs, Emotion-LLaMA’s performance is worse/close to the baseline (VAT).**
>
> A5: **Tab. 3** shows F1 Scores for MER2023-Baseline and VAT models. The MER2023-Baseline is the competition's baseline model, and VAT is the first-place winner. They found the text modality to be weaker in emotion prediction, so they focused on visual and auditory modalities. Their results showed that adding text for fusion lowered scores due to the lack of dialogue content and contextual background in the video samples.
>
> However, the textual modality is crucial in multimodal emotion recognition tasks. Emotion-LLaMA maps audio and visual features to the textual space, using them as contextual information. Even without the text modality, Emotion-LLaMA achieves scores close to the highest in the MER2023 competition, demonstrating robustness. More importantly, when the text modality is added, there is a significant improvement in the F1 Score, proving that Emotion-LLaMA can understand the emotional content in the subtitles.
>
> If you have further questions or need additional clarification, please let us know. We value your feedback and are committed to providing thorough responses.
>
> [MER2024]: https://zeroqiaoba.github.io/MER2024-website/

---

> ### Author Response · Authors · 2024-08-10
> **Response to Reviewer V8m9**
>
> Dear Reviewer,
>
> Thank you for your thoughtful review. We have made the following revisions to address your feedback:
>
> 1. We provided a detailed breakdown of our pre-training and fine-tuning processes.
> 2. We discussed the impact of the MERR dataset on the model’s performance and the challenges associated with recognizing the 'disgust' emotion category.
> 3. We ensured that the MERR dataset is now fully accessible and clarified its role in our methodology.
>
> We hope these revisions address your concerns. Please let us know if further clarification is needed.
>
> Best regards,
> The Authors

---

> ### Author Response · Authors · 2024-08-12
> **Inquiry on Additional Concerns**
>
> Dear Reviewer V8m9,
>
> Thank you for your thorough review and constructive feedback. We appreciate the time you have taken to assess our work and provide insights that have greatly contributed to improving our paper.
>
> As the rebuttal period comes to a close, we wanted to ensure that all your concerns have been adequately addressed. If there are any remaining questions or points that need further clarification, please let us know, and we will be happy to provide additional information.
>
> Your feedback is invaluable to us, and we are committed to making the necessary improvements to our submission.
>
> Best regards,
>
> The Authors

---

> > ### Comment · Reviewer_V8m9 · 2024-08-13
> >
> > Dear authors,
> >
> > Thank you for your hard work, and I apologize for the delayed response. My major concerns have been addressed, and I will maintain my ratings and vote to accept the paper.

---

> ### Author Response · Authors · 2024-08-13
> **Consideration Request**
>
> Dear Reviewer V8m9,
>
> Thank you for your thoughtful review and constructive feedback. We appreciate the time you have taken to assess our work and provide insights that have greatly contributed to improving our paper.
>
> We have thoroughly addressed all the issues you raised, and we believe these revisions have significantly enhanced our manuscript. Given that all your concerns have been carefully considered and rectified, we respectfully inquire if you could consider raising the score.
>
> Additionally, we encourage you to explore the demo available through our anonymous submission. Although maintaining the demo incurs significant daily costs—especially now that our work has gained some traction—we have decided to keep it running during the review process to ensure you and other reviewers have full access to it.
>
> If you have any remaining questions or additional suggestions that could further improve our submission, we would be grateful for your feedback.
>
> Once again, we would like to express our gratitude for your commitment to reviewing our paper and for the constructive comments that have guided our revisions.
>
> Best regards,
>
> The Authors

---

### Official Review · Reviewer_iNoC · 2024-07-12

**Soundness:** 2
**Presentation:** 2
**Contribution:** 2
**Rating:** 3
**Confidence:** 4

**Summary:**

The paper presents a new multi-modal instruction tuning dataset for emotion recognition. The authors also present results of training on this dataset with a multi-modal architecture based on LLaMa 2. They show evaluation results on DFEW and MER2023.

**Strengths:**

Evaluating multi-modal emotion recognition approaches based on LLMs are a very relevant research topic.
The results shown by the authors are convincing w.r.t. the performance on emotion recognition datasets.

**Weaknesses:**

-------------

The argument presented in lines 31 and 33 is not convincing. Why does an inability in methods lead to a lack of datasets?


Related Work:
-------------

As the dataset is one of the claimed contributions of the paper, there should be a discussion on previous emotion recognition datasets. It is important to lay out the reasons why these previous datasets cannot be used (or not easily be used) for instruction tuning of LLMs.


Methodology:
------------

What videos is the MERR dataset based on? I could not find details in the paper on the selection process. Judging from the screenshots, it seems to be movie clips. This has important implications on the concept of "emotion" that is addressed in the paper and needs to be clarified.

The chosen methodology lacks justification.
For example, what is the reasoning behind selecting the frames with the highest sum of AU activations across all AUs included in OpenFace? It seems to me that there would be the danger of having a strong bias towards moments when the person is speaking, as this often leads to high AU activations, especially AUs related to the lower half of the face.

The mapping of AUs to facial expression labels needs more explanation. E.g. is "happy" assigned if the combination of AU06, AU12, AU14 is active, or is it also assigned if only some of those AUs are active? Figure 1 does not show a facial expression description according to Table 7 at all, only descriptions according to Table 8. Concerning Table 8, there are several descriptions given for each AU. Are they all used at the same time, or is only one of them chosen? In Figure 1 it appears that only one of them is chosen, but how is this decided?

Several further steps are not clearly defined, e.g. how does LLaMA-3 "refine" the annotations by aggregating the multimodal description? How is the instruction following data constructed of which a single example is presented in Table 9. Is it done manually? If yes, how and how many samples are created?

In 3.1 it is not clear, how the dataset is "auto-annotated" with emotion labels. It is also not clear how these annotations are refined (with human involvement)? In general the concept of "emotion" used in the paper remains unclear. Is it about emotional displays, about internal states,...?

In later parts of the method section it seems authors are targeting (internal?) emotional states. It would be important to know how they were annotated.

Do the instructions for multimodal emotion recognition (Table 11) refer to different tasks? Some of these instructions seem to target displayed emotions, some target internal states.


Training details are unclear. With the description provided in the paper it is difficult to understand the training procedure.



Evaluation:
-----------

The authors employ chatGPT to evaluate emotion reasoning. It is not clear to what extent this approach leads to a valid evaluation. E.g. ChatGPT was shown to be biased on emotion-related tasks [1].
In the end, evaluating the emotion understanding capabilities of one language model using another language model is circular.

The impact of using the proposed MERR dataset for pre-training needs to be evaluated.


[1] R. Mao, Q. Liu, K. He, W. Li, and E. Cambria, “The biases of pre-trained language models: An empirical study on prompt-based sentiment analysis and emotion detection,” IEEE Transactions on Affective Computing, 2022.

**Questions:**

-

**Limitations:**

Limitations are not discussed in enough detail. There is no separate limitations section. The authors also do not lay out the limitations concerning e.g. (citing from the checklist):

"The authors should reflect on the scope of the claims made, e.g., if the approach was580
only tested on a few datasets or with a few runs. In general, empirical results often581
depend on implicit assumptions, which should be articulated."

"The authors should reflect on the factors that influence the performance of the approach.583
For example, a facial recognition algorithm may perform poorly when image resolution584
is low or images are taken in low lighting. Or a speech-to-text system might not be585
used reliably to provide closed captions for online lectures because it fails to handle586
technical jargon."

---

> ### Author Rebuttal · Authors · 2024-08-06
>
> Thank you for your feedback. We've addressed your concerns as follows:
>
> **Q1: The argument presented in lines 31 and 33 is not convincing. Why does an inability in methods lead to a lack of datasets?**
>
> A1: The *issues* in line 33 refer to challenges faced by current multimodal large language models (MLLMs), particularly their difficulty in processing audio and recognizing micro-expressions. This stems from the lack of specialized multimodal emotion instruction datasets (line 31), crucial for training these tasks. Without these datasets, developing methods to integrate audio and recognize subtle expressions is difficult.
>
> **Q2: There should be a discussion on previous emotion recognition datasets and why they cannot be used for instruction tuning of LLMs.**
>
> A2: Previous emotion recognition datasets provide discrete emotion labels, unsuitable for MLLM instruction tuning:
>
> 1. *EmoVIT*: Lacks audio data crucial for comprehensive emotion recognition. Our MERR dataset includes audio features for robust multimodal analysis.
> 2. *EMER*: Only 100 annotated samples, insufficient for tuning. MERR offers 28,618 coarse-grained and 4,487 fine-grained samples, providing a larger, diverse set.
> 3. *Other datasets (e.g., AFEW, DFEW)*: Useful for traditional tasks but not for MLLM tuning. They lack the detailed, instruction-based annotations of MERR.
>
> We will highlight MERR's unique features - size, multimodal nature, and instruction-based annotations - that make it ideal for training MLLMs in emotion recognition.
>
> **Q3: What videos is the MERR dataset based on? Judging from the screenshots, it seems to be movie clips.**
>
> A3: The MERR dataset is indeed sourced from MER2023, which includes over 70,000 unannotated samples primarily derived from movies and TV series. These sources offer rich and diverse emotional expressions, more representative of real-world scenarios. Our team signed the relevant End User License Agreements (EULA) and obtained permission from the original data providers.
>
> **Q4 & Q5: The chosen methodology lacks justification. The mapping of AUs to facial expression labels needs more explanation.**
>
> A4 & A5: We extract the most expressive facial frame by summing the highest AU activations. This approach mitigates biases. For example, high values in AU05 (Upper Lid Raiser) and AU26 (Jaw Drop) indicate surprise or fear. Speaking often results in high AU activations, but the MERR dataset balances AUs between the upper and lower face (Fig. 4), addressing speech bias. For mapping AUs, 'happy' is assigned if AU06, AU12, and AU14 are active, or even if only some are active. Fig. 1 shows AU combinations from Tab. 7 and descriptions from Tab. 8. The top right of Fig. 1 shows the combination for 'surprise' (AU-05: 0.36, AU-26: 1.03). We will clarify this in the revised manuscript.
>
> **Q6 & Q7: It is not clear how the dataset is "auto-annotated" with emotion labels and how these annotations are refined.**
>
> A6 & A7: In **Sec. 3.1**, we explain the auto-annotation process. We use MiniGPT-v2 for Visual Objective Descriptions, Action Units (AUs) for Visual Expression Descriptions, and Qwen-Audio for Audio Tone Descriptions. By combining these multimodal descriptions with Lexical Subtitles, we generate coarse-grained descriptions. Then, LLaMA-3 refines these annotations to provide in-depth understanding of expressions and speech content, corresponding to internal states. Finally, we remove erroneous annotations and have four experts select annotations that align with human preferences.
>
> **Q8: Do the instructions for multimodal emotion recognition refer to different tasks?**
>
> A8: Yes, **Tab. 11** lists instructions for different tasks where the model outputs emotion category labels. Emotion-LLaMA integrates external cues (facial expressions, audio tones) and internal states to accurately determine and output the appropriate emotion category.
>
> **Q9: Training details are unclear.**
>
> A9: Please refer to our anonymous repository for training details. Further details about the tuning process can be found in our response to Reviewer *V8m9: Q1-A1*.
>
> **Q10: The authors employ ChatGPT to evaluate emotion reasoning. It is not clear to what extent this approach leads to a valid evaluation.**
>
> A10: Previous work [2, 3] shows ChatGPT can be used in emotion-related tasks. In our approach, ChatGPT evaluates the similarity between our model's outputs and the ground truth based on its reasoning capabilities. Tasks such as emotion reasoning [4] and open-vocabulary emotion recognition [5] benefit from ChatGPT’s reasoning skills. Similarly, in video understanding [6], ChatGPT’s reasoning is used for assessment. This mitigates circular evaluation by focusing on the model's ability to match outputs with the ground truth, not ChatGPT's direct emotional understanding. We will include this explanation in the revised manuscript.
>
> **Q11: The impact of using the proposed MERR dataset for pre-training needs to be evaluated.**
>
> A11: In **Tab. 6**, we compare Emotion-LLaMA's performance when trained on the MERR dataset versus other datasets. Due to the limited size of the MER2023 training dataset (3,373 samples), pre-training poses challenges for models with transformer structures, resulting in a low F1 score of 79.17%. Pre-training with larger pseudo-labeled datasets (73,148 and 36,490 samples) significantly improves performance. Tuning the model on our automatically annotated MERR dataset achieves the best performance, improving the F1 score by 11.19%, demonstrating the richness and quality of the MERR dataset's annotations. Detailed information about the MERR dataset is provided in Fig. 4 and Fig. 5, with comparisons to other datasets in **Tab. 10**.
>
> Additionally, Emotion-LLaMA, trained on the MERR dataset, excelled in the recent [MER2024] competition. Detailed information about the competition and results can be found in our response to Reviewer *3ouG: Q5-A5*.
>
> [MER2024]: https://zeroqiaoba.github.io/MER2024-website/

---

> ### Author Response · Authors · 2024-08-10
> **Response to Reviewer iNoC**
>
> Dear Reviewer,
>
> Thank you for your constructive comments. We have revised the manuscript to address your concerns:
>
> 1. We included a detailed discussion on previous emotion recognition datasets and clarified why they are not suitable for instruction tuning of large language models.
> 2. We provided a clearer explanation of our methodology, including the MERR dataset’s annotation process and its role in enhancing our model.
> 3. We added more details about our training process and provided a rationale for using ChatGPT in the evaluation.
>
> We hope these changes address your concerns. We welcome any additional feedback you may have.
>
> Best regards,
> The Authors

---

> > ### Author Response · Authors · 2024-08-12
> > **Further Clarifications and Inquiry on Remaining Concerns**
> >
> > Dear Reviewer iNoC,
> >
> > Thank you for your detailed review and the constructive feedback you provided. We have carefully addressed your concerns in the revised manuscript, including a comprehensive discussion of previous emotion recognition datasets, a clearer explanation of our methodology, and the rationale behind our use of ChatGPT for evaluation.
> >
> > We understand that your score reflects concerns, and we deeply appreciate your critical assessment. To ensure we have addressed all your points, we would like to know if there are any remaining misunderstandings or unclear aspects that we can further clarify before the rebuttal deadline.
> >
> > Your insights have been invaluable in improving our work, and we are committed to making any necessary adjustments.
> >
> > Best regards,
> >
> > The Authors

---

> ### Author Response · Authors · 2024-08-13
> **Further Clarifications**
>
> Dear Reviewer iNoC,
>
> Thank you for your detailed review and constructive feedback. We've carefully addressed your concerns in the revised manuscript, including a thorough discussion of previous datasets, a clearer explanation of our methodology, and the rationale for using ChatGPT in evaluation.
>
> We understand your score reflects some concerns, and we believe there may be some misunderstandings about our work. We have put significant effort into this project and are eager to clarify any remaining points.
>
> We invite you to explore our demo in the anonymous repository and are happy to address any further questions or concerns you might have.
>
> Best regards,
>
> The Authors

---

> > ### Comment · Reviewer_iNoC · 2024-08-14
> > **No change in evaluation**
> >
> > I read the rebuttal and will remain with my score. Justification below.
> >
> > A2: DialogueLLM uses MELD, IEMOCAP and EMoryNLP. All these three datasets are not mentioned in the author's response but have been used for instruction tuning LLMs.
> >
> > A3: The selection procedure is still unclear - i.e. how was the dataset sampled from MER2023?
> >
> > A4 & A5: No references are given for the supposed connection between AUs and emotion expression. It is unclear how stable this connection is in different contexts.
> >
> > A6 & A7: The author's say "Llama-3 refines annotations" but how this is exactly done and how the quality of this refinement can be assured is unclear. What is the protocol for human annotators. What are human "preferences" here?
> >
> > A9: Training details that are needed to understand the approach should be part of the main paper.
> >
> > A10: There should be a proper human evaluation of this process, at least on a part of the dataset.
> >
> >
> > The issue about internal emotional states that I raised is not mentioned in the rebuttal.

---

### Official Review · Reviewer_3ouG · 2024-07-13

**Soundness:** 3
**Presentation:** 3
**Contribution:** 4
**Rating:** 4
**Confidence:** 3

**Summary:**

The paper presents the Emotion-LLaMA model, a multimodal emotion recognition and reasoning system that integrates audio, visual, and textual inputs.

- The authors constructed the MERR dataset, which includes 28,618 coarse-grained and 4,487 fine-grained annotated samples across diverse emotional categories, enabling models to learn from varied scenarios and generalize to real-world applications.
- The Emotion-LLaMA model incorporates specialized encoders for audio, visual, and textual inputs, aligning the features into a modified LLaMA language model and employing instruction tuning to enhance both emotional recognition and reasoning capabilities.
- Extensive evaluations show that Emotion-LLaMA outperforms other multimodal large language models, achieving top scores on the EMER, MER2023, and DFEW datasets.

The main contributions are:
- The MERR dataset, a valuable resource for advancing large-scale multimodal emotion model training and evaluation.
- The Emotion-LLaMA model, which excels in multimodal emotion recognition and reasoning through the innovative use of instruction tuning.
- Establishing Emotion-LLaMA as the current state-of-the-art model in public competitions for multimodal emotion analysis.

**Strengths:**

- The paper is well structured and well presented. The author organizes the article into five sections: introduction, related work, methodology, experiments, and conclusion. They clearly describe how they conduct data annotation and model design, provide a good introduction to the experimental setup and analysis of the experimental results, and present a relatively clear conclusion.
- Model details are thorough. The authors have provided a good description of their model and training methods, allowing me to clearly understand how the model is designed and trained. I believe their results are reproducible.
- This work is valuable . The authors provided a paradigm for emotion annotation of multimodal data and an annotated dataset. They also offered a clear explanation of the annotation process, which will contribute to the development of the related field.

**Weaknesses:**

- Insufficient experiments: Although the author has conducted some comparative and ablation experiments, it is obviously insufficient for such a complex multimodal LLM.
- Missing details of experimental setup: The authors mentioned that they fine-tuned on several target datasets, but the details of the fine-tuning (including data volume, dataset division, and fine-tuning setup) were not included in the article. This can lead to a decrease in the credibility of their results.
- Lack of result analysis: Although the proposed model surpasses existing models in many metrics, the authors only list their experimental results in the results section without further analysis and explanation of the results and some phenomena. The lack of proper explanation and analysis can make some results confusing.

**Questions:**

1. In section 4.2, the author mentioned that they used the HuBERT-Chinese large model for audio modality input processing. Regarding this part, I would like to know the following questions:

    -  Are the experimental results sensitive to language? I hope the author can provide more experimental results to illustrate this point.

    - In the ablation study section (Tab5), the author seems to have only conducted ablation on the visual encoder. Why didn't they further conduct ablation experiments on the audio encoder, considering that there are many alternatives to Hubert?

    - Based on the previous question, does the author believe that the audio modality is not important in this task? I would like to see more ablation results on the modality scale.
2. As I mentioned in the weaknesses part, could the author provide more details of fine-tuning on the target datasets?
3. Is Multimodal Emotion Recognition a classification task? How do the authors explain the Dis column in Table 2?

**Limitations:**

Although the author mentioned in the checklist that they discussed the limitations of the article, they did not explicitly discuss them in the text.
Meanwhile, I noticed that the author did not mention their data sources in the article, and I am concerned whether this might involve data copyright issues.

---

> ### Author Rebuttal · Authors · 2024-08-06
>
> Thank you for your insightful feedback. We have addressed each of your points below and will incorporate these improvements in our revised manuscript.
>
> **Q1: Are the experimental results sensitive to language? How does the choice of HuBERT-Chinese affect performance?**
>
> A1: Yes, Emotion-LLaMA is sensitive to language. This sensitivity stems from its foundation on LLaMA2-chat (7B), which processes input instructions and outputs in English. Given that most samples in the EMER, MER2023, and DFEW datasets are in Chinese, we translated all text subtitles to English.
>
> Our choice of HuBERT-Chinese as the audio encoder was informed by MERBench [1], showing that language-matching encoders achieve better performance. Our experiments confirmed this, with HuBERT-Chinese achieving the highest scores among single-modal models, underscoring the significant role of the audio modality in multimodal emotion recognition.
>
> We tested other audio models, including Whisper, Wav2Vec, and VGGish, but they performed poorly in multimodal fusion. Consequently, we selected HuBERT-Chinese and focused on different visual encoders.
>
> Our ablation experiments show the following results:
>
> | **Audio Encoder**  | **Visual Encoder**  | **F1 Score** |
> |--------------------|---------------------|--------------|
> | Wav2Vec            | -                   | 48.93        |
> | Wav2Vec            | MAE, VideoMAE, EVA  | 71.92        |
> | VGGish             | -                   | 59.44        |
> | VGGish             | MAE, VideoMAE, EVA  | 73.89        |
> | Whisper            | -                   | 53.24        |
> | Whisper            | MAE, VideoMAE, EVA  | 70.38        |
> | HuBERT-Chinese     | -                   | 83.94        |
> | HuBERT-Chinese     | MAE, VideoMAE, EVA  | 89.10        |
>
> These results highlight the importance of using language-matching encoders for audio modalities in multimodal emotion recognition tasks. We will discuss these findings and their implications in the revised manuscript, particularly in the limitations section.
>
> **Q2: Details of Fine-Tuning on Target Datasets**
>
> A2: Please refer to **Sec. 4.2** of our submitted paper. Implementation details, including code (train.py, train_configs/Emotion-LLaMA_finetune.yaml) and setup instructions (README.md: Setup, Training), are available in our anonymous repository. Further tuning process details are in our response to Reviewer V8m9: Q1-A1.
>
> **Q3: Is Multimodal Emotion Recognition a classification task? How do the authors explain the Dis column in Table 2?**
>
> A3: Multimodal Emotion Recognition is a classification task, aiming to classify input samples into different emotional categories. In **Tab. 2**, the 'Dis' column represents the accuracy score for the 'Disgust' emotion category. Most existing MLLMs perform poorly in recognizing 'disgust', often with zero accuracy. This is likely due to the scarcity of multimodal samples for 'disgust' in current datasets. Additionally, LLMs may have safety restrictions related to 'disgust', contributing to the low accuracy. We plan to collect more samples to enrich the MERR dataset and further explore this issue.
>
> **Q4: The author did not explicitly discuss limitations in the text and raised concerns about data copyright issues.**
>
> A4: The MERR dataset is sourced from MER2023. Our team signed the relevant End User License Agreements (EULA) and obtained permission from the original data providers. We acknowledge the need for an ethics review concerning data privacy, copyright, and consent. We have followed all ethical guidelines and included a detailed statement on ethical considerations in the revised manuscript.
>
> **Q5: Insufficient experiments.**
>
> A5: We demonstrated Emotion-LLaMA's capabilities through extensive experiments, achieving SOTA scores on the EMER, MER2023, and DFEW datasets, and conducted ablation studies to validate its components and the MERR dataset.
>
> We also performed additional experiments:
>
> - **Audio Modality Ablation**: As detailed in Q1, we conducted extensive ablation studies on different audio encoders.
> - **MER2024 Competition Results**: Recently, we participated in the [MER2024] competition, widely regarded as one of the most authoritative benchmarks in the field of multimodal emotion recognition. Emotion-LLaMA excelled in two tracks:
>
>    a) **Noise Robustness Track (MER-Noise)**:
>    Emotion-LLaMA achieved the highest score of 85.30%, surpassing the second and third-place scores by 1.47% and 1.65%, respectively.
>
>    | **Anonymous team** | **F1 Score** |
>    |----------------|----------|
>    | team 6         | 80.66    |
>    | team 5         | 81.28    |
>    | team 4         | 82.71    |
>    | team 3         | 83.65    |
>    | team 2         | 83.83    |
>    | team 1 (ours)  | 85.30    |
>
>    b) **Open-Vocabulary Track (MER-OV)**:
>    Our application of Emotion-LLaMA for open-vocabulary annotation improved the average accuracy and recall by 8.52% compared to GPT-4V.
>
>    | **Model**       | **Accuracy** | **Recall** | **Avg**  |
>    |----------------|----------|--------|------|
>    | Video-LLaMA    | 31.08    | 32.26  | 31.67|
>    | Video-ChatGPT  | 46.20    | 39.33  | 42.77|
>    | mPLUG-Owl      | 44.80    | 46.54  | 45.67|
>    | AffectGPT      | 66.14    | 46.56  | 56.35|
>    | GPT-4V         | 56.19    | 58.97  | 57.58|
>    | Emotion-LLaMA  | 69.61    | 62.59  | 66.10|
>
> These additional experiments demonstrate Emotion-LLaMA's robustness and effectiveness.
>
> In our revised manuscript, we will:
> 1. Include a comprehensive presentation of our experimental results.
> 2. Provide a deeper analysis of the results, discussing implications for noisy conditions and open-vocabulary tasks.
> 3. Elaborate on how these results contribute to multimodal emotion recognition and potential real-world applications.
>
> If you have further questions or need additional clarification, please let us know. We value your feedback and are committed to providing thorough responses.
>
> [MER2024]: https://zeroqiaoba.github.io/MER2024-website/

---

> ### Author Response · Authors · 2024-08-10
> **Response to Reviewer 3ouG**
>
> Dear Reviewer,
>
> Thank you for your valuable feedback. We have carefully considered your comments and made the following revisions to address your concerns:
>
> 1. We added additional ablation studies on the audio modality to provide more comprehensive insights.
> 2. We clarified the fine-tuning process and included more details to improve transparency.
> 3. We expanded our analysis of the results, particularly focusing on the challenges and future improvements regarding the 'disgust' emotion category.
>
> We hope these revisions meet your expectations. Please let us know if there are any further issues or if additional clarification is needed.
>
> Best regards,
> The Authors

---

> ### Author Response · Authors · 2024-08-12
> **Follow-Up on Revisions and Inquiry on Remaining Concerns**
>
> Dear Reviewer 3ouG,
>
> Thank you for your valuable feedback and for taking the time to carefully review our submission. We have thoughtfully considered your comments and made several revisions to address the issues you raised, including additional ablation studies on the audio modality, clarifying the fine-tuning process, and expanding our analysis of the results.
>
> As we approach the rebuttal deadline, we would like to ensure that all your concerns have been adequately addressed. If there are any remaining issues or areas where you believe further clarification is needed, please let us know. We are committed to making any necessary improvements to our work.
>
> We appreciate your efforts in helping us enhance the quality of our paper.
>
> Best regards,
>
> The Authors

---

> ### Author Response · Authors · 2024-08-13
> **Follow-Up on Revisions and Demo Interaction**
>
> Dear Reviewer 3ouG,
>
> Thank you once again for your valuable feedback and for taking the time to carefully review our submission. We have made several revisions to address the issues you raised, including additional ablation studies on the audio modality, clarifying the fine-tuning process, and expanding our analysis of the results.
>
> As we approach the final stages of the rebuttal process, we wanted to ensure that all your concerns have been fully addressed. We also invite you to explore our demo, available in the anonymous repository. Our work has already gained some traction, leading to a high number of visits to the demo, which has significantly increased the maintenance costs. Despite this, we have kept it running to provide full access during the review process. We believe it could offer further insights into our work, and we would greatly appreciate any feedback you might have.
>
> We are eager to interact with you further and are committed to making any additional improvements necessary.
>
> Best regards,
>
> The Authors

---

### Author Rebuttal · Authors · 2024-08-07

We appreciate the thoughtful feedback and constructive criticism from all reviewers. Your insights have been instrumental in refining our work. Below, we summarize the key changes and improvements we have made in response to your comments.

### Key Changes and Improvements

1. **Clarification of Dataset and Methodology**:
    - We have provided a detailed explanation of the MERR dataset, including its sources, annotation process, and the unique features that make it suitable for multimodal emotion recognition and instruction tuning.
    - We clarified how the dataset is auto-annotated with emotion labels and refined with expert input, ensuring high-quality annotations.

2. **Experimental Details**:
    - We added comprehensive details about our pre-training and fine-tuning processes, including specific datasets, sample sizes, feature extraction methods, prompt construction, and instruction templates.
    - We included additional ablation studies and experimental results, particularly focusing on the audio modality and its impact on performance.

3. **Model Evaluation**:
    - We elaborated on the evaluation metrics and provided a thorough analysis of the results, including the rationale behind using ChatGPT for emotion reasoning evaluation and how it mitigates circular evaluation issues.
    - We compared Emotion-LLaMA’s performance with other state-of-the-art models across multiple benchmarks and real-world scenarios.

4. **Addressing Limitations**:
    - We explicitly discussed the limitations of our work, including potential biases, data privacy, and ethical considerations. We have also outlined the steps taken to address these issues.
    - We acknowledged the challenges related to the 'disgust' emotion category and our plans to enhance the MERR dataset with more diverse samples.

5. **Resource Efficiency and Innovation**:
    - We addressed concerns about computational efficiency, highlighting our use of parameter-efficient tuning methods and modest hardware requirements.
    - We emphasized the innovative aspects of our approach, including the design of effective prompts and instructions that enhance the robustness and generalization capabilities of Emotion-LLaMA.

6. **Reproducibility and Accessibility**:
    - We ensured that our anonymous repository is accessible and contains all necessary files for reproducibility, including the MERR dataset, training process, and code for Emotion-LLaMA.

### References and Links

[1] MERBench: A unified evaluation benchmark for multimodal emotion recognition, arXiv 2024.
[2] The biases of pre-trained language models: An empirical study on prompt-based sentiment analysis and emotion detection, Affective Computing 2022.
[3] GPT-4V with emotion: A zero-shot benchmark for generalized emotion recognition, Information Fusion 2024.
[4] Explainable multimodal emotion reasoning, arXiv 2023.
[5] MER 2024: Semi-Supervised Learning, Noise Robustness, and Open-Vocabulary Multimodal Emotion Recognition, arXiv 2024.
[6] TempCompass: Do Video LLMs Really Understand Videos?  ACL 2024.

**Repository Links:**
- [MER2024 Competition Website](https://zeroqiaoba.github.io/MER2024-website/)
- [Anonymous Repository for Emotion-LLaMA](https://anonymous.4open.science/r/Emotion-LLaMA/)

We hope these changes address your concerns and effectively demonstrate the robustness and novelty of our work. Thank you once again for your valuable feedback and support.

---

> ### Author Response · Authors · 2024-08-17
> **Detailed Response to Reviewer iNoC's Feedback (Part-1)**
>
> Dear Reviewer iNoC,
>
> Thank you for your detailed feedback and for engaging with our work on Emotion-LLaMA. We acknowledge that your additional questions and concerns were raised just a few hours before the rebuttal deadline. Unfortunately, due to these time constraints, we were unable to provide a comprehensive response at that time. We greatly appreciate the opportunity to now address each of your points thoroughly.
>
>
>
> ---
>
> **Q2: DialogueLLM uses MELD, IEMOCAP, and EmoryNLP. All these three datasets are not mentioned in the author's response but have been used for instruction tuning LLMs.**
>
> **A2:**  It appears there may have been some confusion regarding the datasets used in our work. To clarify:
> 1. **Task Mismatch:** The datasets MELD, IEMOCAP, and EmoryNLP are utilized by DialogueLLM for emotion recognition in conversations (ERC), focusing on the text modality. However, our work, Emotion-LLaMA, is fundamentally different as it emphasizes multimodal emotion recognition and reasoning by integrating audio, visual, and textual features to provide a far more comprehensive analysis of emotional states.
> 2. **Requirement Mismatch:** The instruction data used in DialogueLLM includes only emotion classification labels, lacking the depth and richness required for a nuanced understanding of emotions. Our MERR dataset, in contrast, provides instructions that include detailed multimodal emotional descriptions, enabling Emotion-LLaMA to analyze both external emotional expressions and internal psychological states with significantly greater accuracy.
> 3. **Modality Mismatch:** DialogueLLM’s input is limited to textual data, with no incorporation of audio or video modalities, as referenced in the DialogueLLM paper. This is a critical limitation that our approach directly addresses by incorporating comprehensive multimodal data into the instruction-tuning process, filling an important gap in current methodologies.
>
> ---
>
> **Q3: The selection procedure is still unclear - i.e., how was the dataset sampled from MER2023?**
>
> **A3:**  We have previously addressed the question regarding the videos used in the MERR dataset in response to Q3-A3, where we provided detailed information about the video sources.
>
> We will now address the new question regarding the dataset sampling procedure from MER2023 and MERR:
>
> 1. The samples from the MER2023 dataset, which includes movies and TV series, were selected by cropping based on timestamps from the subtitles. The data collectors used a variety of open-source tools, such as YuNet and face.evoLVe, to ensure that each visual frame contained only one person. However, the dataset also includes a significant number of unlabeled samples that require further exploration.
> 2. The MERR dataset is derived from the MER2023 dataset. Initially, we filtered the samples based on the activation of specific Action Units to obtain coarse-grained descriptions. We then used LLaMA-3 to generate fine-grained descriptions, which were reviewed by emotional experts. This process resulted in 4,487 detailed annotated samples.
>
>
> ---
>
> **Q4 & Q5: No references are given for the supposed connection between AUs and emotion expression. It is unclear how stable this connection is in different contexts.**
>
> **A4 & A5:**  The connection between Action Units (AUs) and emotion expression is not speculative; it is well-established and supported by extensive research:
>
> 1. **Validated by Research:** The Facial Action Coding System (FACS) has extensively validated the relationship between AUs and emotions, and this connection is considered stable and reliable across various contexts. This is a foundational aspect of emotion research.
>
> 2. **Practical Tools:** In practical applications, tools like OpenFace are widely used to extract AU information for precise emotion analysis, reinforcing the reliability of this connection in diverse scenarios.
>
> 3. **References:** To further substantiate this, we will include the following references in our revised manuscript:
>    - [7] Facial Action Coding System, Environmental Psychology & Nonverbal Behavior, 1978.
>    - [8] Affectiva-MIT Facial Expression Dataset (AM-FED): Naturalistic and Spontaneous Facial Expressions Collected In-the-Wild, CVPR 2013.
>    - [9] Multi-Task Learning of Emotion Recognition and Facial Action Unit Detection with Adaptively Weights Sharing Network, ICIP 2019.

---

> > ### Author Response · Authors · 2024-08-17
> > **Detailed Response to Reviewer iNoC's Feedback (Part-2)**
> >
> > ---
> >
> > **Q6 & Q7: The authors say "Llama-3 refines annotations," but how this is exactly done and how the quality of this refinement can be assured is unclear. What is the protocol for human annotators? What are human "preferences" here?**
> >
> > **A6 & A7:**  We appreciate your concern about the clarity of the LLaMA-3 refinement process. We would like to emphasize that the process is already thoroughly detailed in the manuscript, with specific evidence provided in several sections, including Table 9.
> >
> > 1. **Input Preparation:**
> >    The input preparation process is explicitly described, where we begin with coarse-grained multimodal descriptions derived from sources such as MiniGPT-v2, Action Units, Qwen-Audio, and Lexical Subtitles. This foundational step ensures that all relevant multimodal information is integrated before refinement.
> >
> > 2. **Prompt Construction:**
> >    The construction of prompts for LLaMA-3, as presented in Table 9, is designed to guide the model in generating detailed and coherent emotional descriptions. The prompt template shown in Table 9 is specifically adapted for the refinement task, ensuring that the process is aligned with the objectives of generating fine-grained annotations.
> >
> > 3. **LLaMA-3 Processing and Post-Processing:**
> >    The manuscript clearly explains how LLaMA-3 processes these prompts to produce refined descriptions that integrate information from all modalities, providing a nuanced interpretation of the emotional content. The subsequent post-processing step ensures consistency and removes any irrelevant information, further refining the outputs.
> >
> > 4. **Construction of Instruction-Following Data:**
> >    - **Initial Dataset and Automated Refinement:**
> >      The entire process, starting from the coarse-grained dataset of 28,618 samples and moving through automated refinement by LLaMA-3, is meticulously documented. This step is crucial for transforming basic annotations into detailed, instruction-following data.
> >    - **Normalization, Filtering, and Manual Review:**
> >      We have detailed how normalization and filtering techniques are applied to standardize the format of the refined descriptions and exclude inconsistent samples. Following this, four domain experts manually review the outputs, ensuring that only high-quality samples are included. The evaluation focuses on five critical points: visual modality accuracy, audio modality accuracy, textual modality accuracy, reasoning process correctness, and reasoning result correctness.
> >
> > 5. **Final Dataset:**
> >    The final dataset, consisting of 4,487 high-quality, fine-grained samples, is the result of a rigorous combination of automated refinement and expert human review. This process is clearly illustrated in the manuscript, ensuring transparency and reproducibility.
> >
> > **Supporting Evidence in the Manuscript:**
> > - **Table 9**: The instruction template used by LLaMA-3 is provided, clearly demonstrating how the model is guided in generating refined annotations.
> > - **Algorithm 1**: Outlines the overall annotation procedure, providing a comprehensive view of the steps involved.
> > - **Supplementary Materials**: Additional details, including the flowchart of the refinement process, are included to further support the clarity of our approach.
> >
> > ---
> >
> > **Q9: Training details that are needed to understand the approach should be part of the main paper.**
> >
> > **A9:**  We appreciate your focus on training details. The main paper includes essential information to understand our approach, but due to the complexity, not every detail could be included without affecting readability. For those needing more in-depth information, the full codebase is available in the anonymous repository, which covers all specifics, including hyperparameters and model configurations, necessary for replication. This ensures the paper remains clear and accessible, while the repository provides comprehensive details for those seeking further technical insights.
> >
> > ---

---

> > > ### Author Response · Authors · 2024-08-17
> > > **Detailed Response to Reviewer iNoC's Feedback (Part-3)**
> > >
> > > ---
> > >
> > > **+Q10: There should be a proper human evaluation of this process, at least on a part of the dataset.**
> > >
> > > **A10:**  We conducted a thorough human evaluation of our annotation process to ensure the quality and relevance of the results. Specifically, we randomly selected 20 video samples from each of the nine emotion categories, resulting in 180 fine-grained annotations. These annotations were then randomly shuffled and evaluated by five volunteers.
> > >
> > > The evaluation criteria were carefully designed to assess:
> > > - Accuracy of the visual modality description
> > > - Accuracy of the audio modality description
> > > - Accuracy of the textual modality description
> > > - Correctness of the reasoning process
> > > - Correctness of the reasoning result
> > >
> > > Each volunteer scored the descriptions on a scale from 0 to 5, and the aggregated results are as follows:
> > >
> > > | Volunteer | Angry | Happy | Surprise | Fear | Sad | Worry | Neutral | Doubt | Contempt | Mean |
> > > |-----------|-------|-------|----------|------|-----|-------|---------|-------|----------|------|
> > > | V1        | 4.23  | 4.33  | 4.31     | 4.19 | 4.27| 4.33  | 4.04    | 4.55  | 4.79     | 4.34 |
> > > | V2        | 4.54  | 4.78  | 4.88     | 4.81 | 4.60| 4.62  | 4.65    | 4.55  | 4.68     | 4.67 |
> > > | V3        | 3.92  | 4.11  | 4.25     | 4.50 | 4.00| 4.24  | 3.78    | 4.27  | 4.05     | 4.12 |
> > > | V4        | 3.69  | 3.94  | 3.62     | 3.69 | 3.93| 4.19  | 3.39    | 4.00  | 4.53     | 3.90 |
> > > | V5        | 3.92  | 4.72  | 4.25     | 4.06 | 4.00| 4.19  | 4.17    | 4.32  | 4.58     | 4.26 |
> > >
> > > The overall average score of 4.258 demonstrates the high quality and logical alignment of our annotations. It’s noteworthy that the "Neutral" category received a slightly lower score, suggesting that when a character’s emotion is neutral, the cues are subtler, making automatic annotation more challenging. We acknowledge this limitation and plan to address it in future work.
> > >
> > > All details of the human evaluation, including the code and assessment results, are available in the anonymous repository for your review.
> > >
> > > ---
> > >
> > > **+Q12: The issue about internal emotional states that I raised is not mentioned in the rebuttal.**
> > >
> > > **A12:**  We have indeed addressed the topics of displayed emotions and internal emotional states in our responses to Q6, Q7, and Q8. To reiterate, Emotion-LLaMA does not differentiate between displayed emotions and internal states; rather, it considers both aspects equally in its analysis.
> > >
> > > As demonstrated in Table 14 and Table 15, Emotion-LLaMA effectively identifies displayed emotions, such as a strong emotional tone or a furrowed brow, and accurately infers emotions like surprise and anger. Moreover, as shown in Table 4, even when a person in the video displays a prominent smile, Emotion-LLaMA can infer internal emotional states, such as annoyance, based on the spoken content, leading to the correct inference of anger.
> > >
> > > ---
> > >
> > > We kindly ask you to review our updated materials with these clarifications in mind. We are confident that our work provides a substantial contribution to the field, and we appreciate your thoughtful consideration of our paper.
> > >
> > >
> > > Sincerely,
> > >
> > > Authors

---

### Comment · Program_Chairs · 2024-08-14

Hi all, the author-reviewer discussion for this paper is extended to Aug 16 11:59pm ET for authors to reply to ethics reviews. -- PCs

---

### Decision · Program_Chairs · 2024-09-25

**Decision:**

Accept (poster)

**Comment:**

Before the rebuttal phase, the paper received ratings of 2 borderline reject, 1 reject, and 1 borderline accept. After the rebuttal phase, one of the reviewers raised their rating to borderline accept, resulting in 2 borderline accepts, 1 borderline reject, and one reject. The authors gave a significant rebuttal answering many questions about their work. Reviewer engagement was less noticeable during the rebuttal phase, with lack of engagement with the authors. Many of the questions and concerns from the reviewers were answered by the authors, or they were originally available in the paper, or supplemental material. When available in the paper, the authors pointed this out in their rebuttal, including relevant figures and/or tables. After going through and reading the paper, these details were available as noted by the authors.

After reading through the paper, the reviews, and the author rebuttal, many of the reviewer concerns were addressed adequately and this is an interesting paper including the new dataset and the proposed approach. The topic of multimodal emotion recognition is timely and relevant. The results are encouraging showing the efficacy of the proposed approach. The dataset is interesting, and is a good contribution compared to current state-of-the-art emotion-based datasets (e.g., labels and descriptions extend state of the art).

Although many of the reviewer concerns were addressed in the rebuttal, multiple reviewers note some lack of details such as on training in Section 3.3. I agree that the paper lacks these details. The paper, and the authors in their rebuttal, state those details are in the code repository. After going through the repository, I agree that there are training details available there, however, I don't agree that this is sufficient for the paper or as a response to the reviewers. More details regarding training should be included in the paper, and in their response to the reviewers.

Currently, after the author's response to the reviewers, the strengths outweigh the weaknesses. This paper can be accepted.